# CovMatch: Cross-Covariance Guided Multimodal Dataset Distillation with Trainable Text Encoder

**Yongmin Lee**
School of Electrical Engineering
KAIST
lym7505@kaist.ac.kr

**Hye Won Chung**
School of Electrical Engineering
KAIST
hwchung@kaist.ac.kr

## Abstract

Multimodal dataset distillation aims to synthesize a small set of image-text pairs that enables efficient training of large-scale vision-language models. While dataset distillation has shown promise in unimodal tasks, extending it to multimodal contrastive learning presents key challenges: learning cross-modal alignment and managing the high computational cost of large encoders. Prior approaches address scalability by freezing the text encoder and updating only the image encoder and text projection layer. However, we find this severely limits semantic alignment and becomes a bottleneck for performance scaling. We propose Cov-Match, a scalable dataset distillation framework that aligns the cross-covariances of real and synthetic features while regularizing feature distributions within each modality. Unlike prior approaches, CovMatch enables joint optimization of both encoders, leading to stronger cross-modal alignment and improved performance. Evaluated on Flickr30K and COCO, CovMatch outperforms state-of-the-art multimodal distillation methods and achieves up to 6.8% absolute gains in retrieval accuracy using only 500 synthetic pairs. Our code is available at https://github.com/Yongalls/CovMatch.

## 1 Introduction

Dataset distillation aims to synthesize a compact and representative dataset from a much larger one, enabling efficient model training with significantly reduced computational cost. While this approach has shown strong results in unimodal settings, particularly in image classification, its extension to multimodal tasks remains underexplored. The success of large-scale multimodal models such as CLIP [36] has relied on training with massive image-text datasets, often comprising hundreds of millions of examples, which poses serious computational and storage challenges. This gap motivates the study of multimodal dataset distillation, which seeks to generate a small, high-quality set of image-text examples that enables efficient multimodal training. However, distillation in the multimodal setting introduces new challenges that are both algorithmic and computational.

A central difficulty in vision-language distillation lies in learning accurate cross-modal correspondences between image and text, which typically involves powerful pretrained encoders and contrastive learning frameworks. Incorporating such setups into a distillation pipeline significantly amplifies the computational and memory burden, especially under the standard bi-level optimization framework of dataset distillation [41], where the inner loop updates model parameters using the synthetic data, and the outer loop updates the synthetic dataset based on the model's performance on real data. Computing gradients for the outer loop requires tracking how model parameters evolve with respect to synthetic data, often necessitating full unrolling of the inner loop, which is expensive even in unimodal settings. In multimodal contrastive learning, scalability is even more constrained, as it involves significantly larger models, such as NFNet (140MB) and BERT (450MB), compared to the lightweight ConvNets (e.g., 1.24MB) commonly used in unimodal distillation.

39th Conference on Neural Information Processing Systems (NeurIPS 2025).

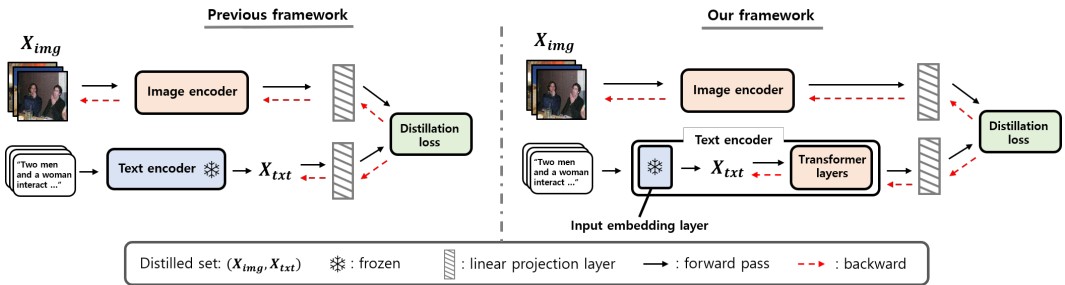

Figure 1: Comparison of multimodal dataset distillation frameworks. Prior methods freeze the entire text encoder and pass distilled text only through a linear projection. In contrast, we freeze only the input embedding layer (i.e., token and position embedding module) and include the transformer layers in the distillation process.

Table 1: **Resource requirements for long-term trajectory matching methods on a single A100 80GB GPU.** For the unimodal case, we apply MTT [5] to distill CIFAR-10 using a 1.24MB ConvNet. In the multimodal setting, we distill COCO [26] using NFNet (140MB) and BERT (450MB) as encoders, applying MTT and Tesla [8] by matching trajectories for both modalities. The cost of storing expert trajectories scales with model size and checkpoint count, while distillation cost scales with model complexity, synthetic steps, and synthetic data batch size $|\mathcal{B}^{\text{syn}}|$. Despite using reduced hyperparameters, long-term matching in the multimodal case remains costly. In contrast, CovMatch requires no expert trajectories and achieves distillation with much lower cost, scaling only with model complexity and the batch sizes $|\mathcal{B}^{\text{syn}}|$ and $|\mathcal{B}^{\text{real}}|$.

| Method | Hyper-parameters | | | | Expert | | Distill | |
| --- | --- | --- | --- | --- | --- | --- | --- | --- |
| | checkpoints | syn steps | $|\mathcal{B}^{\text{syn}}|$ | $|\mathcal{B}^{\text{real}}|$ | storage | time | memory | time |
| MTT (unimodal) | 5000 | 50 | 128 | - | 6.2GB | 1.9h | 15GB | 1.0 sec/it |
| MTT (multimodal) | 200 | 8 | 20 | - | 120GB | 132h | 71GB | 16.9 sec/it |
| Tesla (multimodal) | 200 | 8 | 20 | - | 120GB | 132h | 22GB | 19.2 sec/it |
| CovMatch | - | - | 100 | 128 | 0GB | 0h | 15GB | 1.2 sec/it |

To address scalability issues, initial multimodal distillation methods such as MTT-VL [43] and LoRS [45] adopt Matching Training Trajectories (MTT) [5] as a surrogate for bi-level optimization, while freezing the text encoder and updating only the image encoder and the projection layers. MTT precomputes and stores expert training trajectories from real data, and optimizes the synthetic dataset to match segments of these trajectories. While MTT avoids full unrolling of the inner optimization loop, it still requires storing expert checkpoints and unrolling gradients with respect to the synthetic data to align with the expert trajectories. Applying MTT to both image and text encoders in multimodal setups exacerbates scalability issues, as summarized in Table 1. For instance, storing expert trajectories for large backbones such as NFNet (image encoder) and BERT (text encoder) can require over 120GB of storage and 5 days of training on a single A100 GPU. Moreover, synthesizing just 100 image-text pairs for COCO dataset [26] requires over 70GB of memory for distillation, necessitating the use of high-end GPUs like the A100 80GB. Even memory-efficient variants like Tesla [8] remain costly due to expert trajectory storage. As a result, current approaches typically freeze the text encoder, as shown in Figure 1(left), and synthesize image-text pairs that update only the image encoder and the image/text projection layers.

However, we find that freezing the text encoder and relying solely on the projection layer for aligning modalities severely limits the capacity for semantic alignment in multimodal contrastive learning. In particular, this design becomes a bottleneck in scaling the performance of dataset distillation. In our analysis, we train vision-language models using LoRS-generated synthetic image-text pairs [45] and observe that captions corresponding to the same image in the original Flickr30K dataset [35] fail to form coherent clusters in the text embedding space, even as the size of the synthetic dataset increases (Figure 2(a)). This indicates that a frozen text encoder is insufficient to support the semantic alignment necessary for effective cross-modal learning. As a result, image-text retrieval performance with the LoRS saturates with increasing synthetic data size and, beyond $N = 1000$ image-text pairs, underperforms models trained on randomly sampled real pairs (Figure 2(c)).

Motivated by these observations, we propose a new vision-language dataset distillation framework (Figure 1(right)) that synthesizes image-text pairs to update both image and text encoders, enabling stronger cross-modal alignment while remaining scalable even with large pretrained models. The core idea is to simplify the inner optimization of the bi-level distillation framework into a closed-form solution, thereby avoiding the high memory and computational cost associated with unrolled optimization or trajectory matching. Inspired by prior works in the unimodal setting such as KIP [32] and FrePo [54], we fix the encoders during each distillation step and optimize only the linear projection layers. By adopting the linear multimodal contrastive loss [31], the inner optimization admits a closed-form solution, reducing the bi-level objective to maximizing the trace of inner product between the cross-covariance matrices of real and synthetic image-text features. This enables efficient outer optimization without unrolling, as gradients are simply backpropagated through the fixed encoders.

Building on this insight, we introduce Cross-Covariance Matching (CovMatch), an efficient algorithm for multimodal dataset distillation. CovMatch aligns cross-covariance matrices between real and synthetic pairs sampled at each distillation step, and includes a regularization term that matches feature distributions within each modality to prevent trivial solutions. To ensure the synthetic data remains informative for encoder training, we incorporate a lightweight online model update, where the encoders are updated using a small batch of real image-text pairs before each distillation step, keeping the alignment statistics in sync with the evolving representations.

We evaluate CovMatch on image-text retrieval tasks using the Flickr30K [35] and COCO [26] benchmarks. CovMatch consistently outperforms state-of-the-art multimodal distillation methods, including MTT-VL [43] and LoRS [45]. Remarkably, with just 500 synthetic pairs, CovMatch achieves absolute improvements of 6.8% on Flickr30K and 6.1% on COCO in average retrieval accuracy over the best-performing baseline. These gains are largely attributed to CovMatch's ability to jointly optimize both image and text encoders without compromising scalability.

## 2 Motivation

### 2.1 Preliminaries

We introduce the problem of multimodal dataset distillation for image-text contrastive learning and briefly summarize existing approaches. A detailed review of related works is provided in Appendix A.

**Image-Text Contrastive Learning**  Image-text contrastive learning aims to map visual and textual data into a shared embedding space using a bidirectional contrastive loss [36]. Given a dataset of paired image-text samples $\mathcal{T} = \{(x_v^i, x_l^i)\}_{i=1}^M$, where $x_v^i \in \mathbb{R}^v$ and $x_l^i \in \mathbb{R}^l$ denote the $i$-th image and text inputs, the goal is to train an image encoder $f_v : \mathbb{R}^v \to \mathbb{R}^{d_v}$ and a text encoder $f_l : \mathbb{R}^l \to \mathbb{R}^{d_l}$, each followed by a trainable linear projection layer, $G_v \in \mathbb{R}^{z \times d_v}$ and $G_l \in \mathbb{R}^{z \times d_l}$. These projection layers map the modality-specific embeddings into a shared space $\mathbb{R}^z$ that captures semantic similarity. For a given pair $(x_v, x_l)$, the encoders produce intermediate representations $h_v = f_v(x_v; \theta_v)$ and $h_l = f_l(x_l; \theta_l)$, which are then projected as $z_v = G_v h_v$ and $z_l = G_l h_l$. The similarity between image $x_v^i$ and text $x_l^j$ is computed using cosine similarity, $s_{ij} := \cos(z_v^i, z_l^j) = \langle z_v^i, z_l^j \rangle / \|z_v^i\| \|z_l^j\|$ in the shared embedding. The model is trained using the InfoNCE loss [34] with temperature $\tau > 0$:

$$\mathcal{L}_{\text{NCE}} = -\frac{1}{M} \sum_{i=1}^M \left[ \log \frac{\exp(s_{ii}/\tau)}{\sum_{j \neq i} \exp(s_{ij}/\tau)} + \log \frac{\exp(s_{ii}/\tau)}{\sum_{j \neq i} \exp(s_{ji}/\tau)} \right]. \tag{1}$$

**Main Goal of Vision-Language Dataset Distillation**  The goal of vision-language dataset distillation is to compress a large dataset $\mathcal{T} = \{(x_v^i, x_l^i)\}_{i=1}^M$ into a much smaller synthetic dataset $\mathcal{S} = \{(\hat{x}_v^i, \hat{x}_l^i)\}_{i=1}^N$ with $N \ll M$, such that a model trained on $\mathcal{S}$ achieves comparable image-text alignment performance to one trained on $\mathcal{T}$. Following prior work [43, 45], we evaluate $\mathcal{S}$ using standard image-to-text and text-to-image retrieval metrics. Let $d : \mathbb{R}^z \times \mathbb{R}^z \to \mathbb{R}$ denote a similarity metric in the shared embedding space. The image and text representations are computed as $z_v(\theta_v, G_v) = G_v f_v(x_v; \theta_v)$ and $z_l(\theta_l, G_l) = G_l f_l(x_l; \theta_l)$, given model parameters $\Theta = (\theta_v, \theta_l, G_v, G_l)$. Given a test set $\mathcal{D}^{\text{test}}$, the objective is to ensure that the expected similarity remains consistent between models trained on the full dataset $\mathcal{T}$ and on the synthetic dataset $\mathcal{S}$:

$$\mathbb{E}_{(x_v, x_l) \sim \mathcal{D}^{\text{test}}}[d(z_v(\theta_v^*, G_v^*), z_l(\theta_l^*, G_l^*))] \simeq \mathbb{E}_{(x_v, x_l) \sim \mathcal{D}^{\text{test}}}[d(z_v(\hat{\theta}_v, \hat{G}_v), z_l(\hat{\theta}_l, \hat{G}_l))], \tag{2}$$

where $\Theta^* = (\theta_v^*, \theta_l^*, G_v^*, G_l^*)$ and $\hat{\Theta} = (\hat{\theta}_v, \hat{\theta}_l, \hat{G}_v, \hat{G}_l)$ are the parameters trained on $\mathcal{T}$ and $\mathcal{S}$, resp.

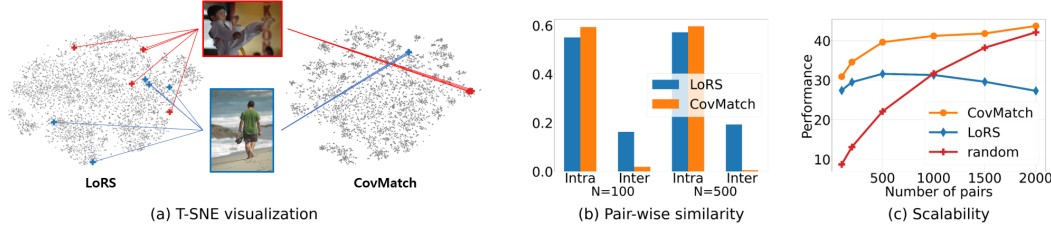

| (a) T-SNE visualization | (b) Pair-wise similarity | (c) Scalability |

Figure 2: (a) T-SNE visualization of text embeddings from the Flickr30K test set. Models are trained with $N = 500$ synthetic pairs distilled by LoRS (left) and CovMatch (right). (b) Intra- and inter-pair average similarity of test captions for models trained with $N = 100$ and $500$ synthetic samples. (c) Retrieval performance as a function of the number of synthetic pairs $N$.

**Bi-Trajectory Matching**  As an initial approach to vision-language dataset distillation, recent works such as MTT-VL and LoRS [43, 45] build on Matching Training Trajectories (MTT) [5], originally proposed for image classification. MTT constructs synthetic datasets by matching the training trajectories of models trained on real data. To extend this to the multimodal setting, MTT-VL introduces bi-trajectory matching, where both image and text encoders are jointly trained using the contrastive loss (1), and the synthetic dataset is optimized to reproduce their joint training dynamics. Let $\tau^* = \{\Theta_t^*\}_{t=0}^T$ denote the expert trajectory with model parameters $\Theta_t^* = (\Theta_{\text{img},t}^*, \Theta_{\text{txt},t}^*)$ obtained from training on the real dataset $\mathcal{T}$. A student model is initialized at a random epoch $s$ with $\hat{\Theta}_s = \Theta_s^*$ and trained on synthetic data $\mathcal{S}$ for $R'$ steps to yield $\hat{\Theta}_{s+R'}$. The synthetic dataset is then updated by minimizing the bi-trajectory matching loss:

$$\mathcal{L}_{\text{bi-trajectory}} = \frac{\|\hat{\Theta}_{\text{img},s+R'} - \Theta_{\text{img},s+R}^*\|_2^2}{\|\Theta_{\text{img},s}^* - \Theta_{\text{img},s+R}^*\|_2^2} + \frac{\|\hat{\Theta}_{\text{txt},s+R'} - \Theta_{\text{txt},s+R}^*\|_2^2}{\|\Theta_{\text{txt},s}^* - \Theta_{\text{txt},s+R}^*\|_2^2}, \tag{3}$$

where $R$ is a target step for matching. This objective encourages the model trained on $\mathcal{S}$ to follow the update trajectory of the model trained on real data. However, computing this loss requires unrolling $R'$ steps of gradient descent for both encoders, leading to significant memory and compute overhead (Table 1). To mitigate this, prior work freezes the pre-trained text encoder (e.g., BERT) and updates only its projection layer $G_l$. As a result, only $\Theta_{\text{txt},t} = G_{l,t}$ is trainable on the text side, while $\Theta_{\text{img},t} = (\theta_{v,t}, G_{v,t})$ remains trainable. We discuss the limitations of this partial freezing strategy.

## 2.2 Limitations of Bi-Trajectory Matching Methods with a Frozen Text Encoder

We find that freezing the text encoder and updating only the projection layer for modality alignment severely limits semantic alignment in multimodal contrastive learning, creating a bottleneck for scaling the performance of dataset distillation. To investigate this, we train vision-language models using synthetic image-text pairs generated by LoRS [45], which freezes the text encoder during distillation. We visualize the resulting text embeddings from the Flickr30K test set [35] in Figure 2(a). Captions associated with the same image (red for the top image, blue for the bottom) fail to form tight clusters in the shared embedding space, suggesting that the frozen text encoder cannot adapt to visual context. In contrast, our method, CovMatch, which jointly optimizes both encoders, yields more semantically coherent text embeddings that better reflect shared visual semantics.

To quantify this observation, we measure the average intra- and inter-pair cosine similarities between text embeddings. Intra-similarity is computed among captions corresponding to the same image, while inter-similarity is computed among captions of different images. As shown in Figure 2(b), LoRS yields a relatively small gap between intra- and inter-similarity, whereas CovMatch produces a significantly larger gap, indicating stronger semantic alignment. This pattern remains consistent as the number of synthetic samples increases from $N = 100$ to $N = 500$.

This limited alignment in LoRS leads to poor performance scaling: as shown in Figure 2(c), retrieval accuracy saturates or even degrades beyond $N = 1,000$ synthetic pairs, eventually falling below models trained on randomly sampled real pairs. In contrast, CovMatch maintains steady performance gains with more synthetic data, enabled by its efficient cross-covariance alignment objective. This scalable formulation allows the synthetic dataset to effectively support the joint training of both encoders, a key factor in its superior alignment and retrieval performance.

## 3 Method

### 3.1 Cross-Covariance Alignment: Closed-Form Solution for Linearized Contrastive Learning

To develop a compute-efficient multimodal dataset distillation method that updates both image and text encoders, we revisit the original formulation of dataset distillation. In the context of image-text contrastive learning, the distillation objective can be framed as a bi-level optimization problem:

$$\mathcal{S}^* = \arg\min_{\mathcal{S}} \mathcal{L}_{\text{NCE}}(\hat{\Theta}; \mathcal{T}) \quad \text{where} \quad \hat{\Theta} = \arg\min_{\Theta} \mathcal{L}_{\text{NCE}}(\Theta; \mathcal{S}). \tag{4}$$

Here, the inner loop learns model parameters $\hat{\Theta}$ from the synthetic dataset $\mathcal{S}$, while the outer loop updates $\mathcal{S}$ to improve performance on the real dataset $\mathcal{T}$. However, computing gradients of the outer loss with respect to $\mathcal{S}$ requires backpropagating through the entire inner optimization, which is costly in both memory and computation. To mitigate this, prior work in unimodal distillation has proposed simplifying the inner loop to admit a closed-form solution. For instance, FrePo [54] considers training only the final linear layer while fixing the feature extractor. Under this setting, the outer objective can be approximated as a kernel ridge regression problem based on the Gram matrix of neural features from real and synthetic inputs. This allows the synthetic dataset to be updated by backpropagating only through the conjugate kernel and fixed feature extractor, greatly reducing the computational cost.

Following this insight, we propose a distillation framework for image-text contrastive learning in which the inner optimization admits a closed-form solution, enabling direct gradient updates to the synthetic dataset $\mathcal{S}$. To achieve this, we fix the image and text encoders $f_v(\cdot; \theta_v)$ and $f_l(\cdot; \theta_l)$ and optimize only the linear projection layers $G_v$ and $G_l$ at each distillation step. Given an input pair $(x_v, x_l)$, we extract features $h_v = f_v(x_v; \theta_v)$ and $h_l = f_l(x_l; \theta_l)$, and project them to a shared embedding space as $z_v = G_v h_v$ and $z_l = G_l h_l$. We adopt the linear contrastive loss from [31, 15]:

$$\mathcal{L}_{\text{lin}}(G_v, G_l; \mathcal{D}) = \frac{1}{2|\mathcal{D}|(|\mathcal{D}|-1)} \sum_{i=1}^{|\mathcal{D}|} \sum_{j \neq i} (s_{ij} - s_{ii}) + \frac{1}{2|\mathcal{D}|(|\mathcal{D}|-1)} \sum_{i=1}^{|\mathcal{D}|} \sum_{j \neq i} (s_{ji} - s_{ii}) + \frac{\rho}{2} \|G_v^\top G_l\|_F^2, \tag{5}$$

where $s_{ij} := (G_v h_v^i)^\top (G_l h_l^j)$ and $\mathcal{D} = \{(h_v^i, h_l^i)\}_{i=1}^{|\mathcal{D}|}$. This loss is equivalent to a trace-based cross-covariance formulation::

$$\mathcal{L}_{\text{lin}}(G_v, G_l; \mathcal{D}) = -\operatorname{Tr}(G_v C^{\mathcal{D}} G_l^\top) + \frac{\rho}{2} \|G_v^\top G_l\|_F^2, \tag{6}$$

where the cross-covariance matrix $C^{\mathcal{D}}$ is given by

$$C^{\mathcal{D}} = \frac{1}{|\mathcal{D}|-1} \sum_{i=1}^{|\mathcal{D}|} (h_v^i - \mu_{h_v})(h_l^i - \mu_{h_l})^\top, \tag{7}$$

with $\mu_{h_v}$ and $\mu_{h_l}$ denoting the empirical means of the image and text features, respectively.

Under this setup, the dataset distillation objective can be written as

$$\mathcal{S}^* = \arg\min_{\mathcal{S}} L_{\text{lin}}(\hat{G}_v, \hat{G}_l; \mathcal{T}) \text{ where } \hat{G}_v, \hat{G}_l = \arg\min_{G_v, G_l} L_{\text{lin}}(G_v, G_l; S), \tag{8}$$

which is equivalent to

$$\mathcal{S}^* = \arg\min_{\mathcal{S}} -\operatorname{Tr}(\hat{G}_v C^{\mathcal{T}} \hat{G}_l^\top) \text{ where } \hat{G}_v, \hat{G}_l = \arg\min_{G_v, G_l} -\operatorname{Tr}(G_v C^{\mathcal{S}} G_l^\top) + \frac{\rho}{2} \|G_v^\top G_l\|_F^2, \tag{9}$$

with $C^{\mathcal{T}}$ and $C^{\mathcal{S}}$ denoting the cross-covariance matrices of the real and synthetic datasets, respectively. The inner loss further admits the reformulation:

$$-\operatorname{Tr}(G_v C^{\mathcal{S}} G_l^\top) + \frac{\rho}{2} \|G_v^\top G_l\|_F^2 = \frac{\rho}{2} \|G_v^\top G_l - \frac{1}{\rho} C^{\mathcal{S}}\|_F^2 - \frac{1}{2\rho} \|C^{\mathcal{S}}\|_F^2,$$

where the optimal solution satisfies $\hat{G}_v^\top \hat{G}_l = \frac{1}{\rho} C^{\mathcal{S}}$. Substituting this into the outer loss in (9) yields

$$\mathcal{S}^* = \arg\max_{\mathcal{S}} \operatorname{Tr}(C^{\mathcal{T}^\top} C^{\mathcal{S}}), \tag{10}$$

indicating that the optimization reduces to aligning the cross-covariances of real and synthetic data. We provide theoretical justification for linear contrastive loss (5) and complete derivation in Appendix C.

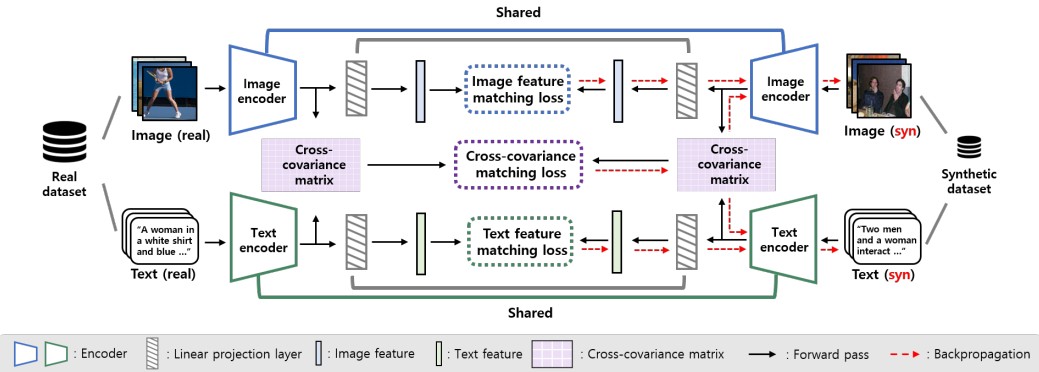

Figure 3: **Illustration of our proposed method, CovMatch.** At each distillation step, CovMatch updates the synthetic image-text pairs using a matching loss computed with an alternately updated online model. The loss consists of two components: 1) Cross-covariance matching loss (11), computed as the distance between the cross-covariance matrices of real and synthetic features (extracted before the projection layers); and 2) Feature matching loss (12), computed as the distance between the mean feature vectors of real and synthetic data after the projection, applied independently to each modality.

## 3.2 CovMatch: Cross-Covariance Matching Algorithm

**Cross-Covariance Matching Loss** As shown in (10), multimodal dataset distillation for linear contrastive learning can be reformulated as aligning the cross-covariance matrices of real and synthetic datasets. However, directly optimizing the trace objective in (10) can lead to training instability, as the trace is unbounded. To mitigate this, we instead frame the objective as a Frobenius norm minimization between the real cross-covariance $C^{\mathcal{T}}$ and the ideal projection-based approximation $\hat{G}_v^\top \hat{G}_l = \frac{1}{\rho} C^{\mathcal{S}}$:

$$\mathcal{L}^{\text{cov}}(\mathcal{T}, \mathcal{S}) = \|\rho \cdot C^{\mathcal{T}} - C^{\mathcal{S}}\|_F^2. \tag{11}$$

Here, the hyperparameter $\rho$ adjusts the scale differences between real and synthetic data, possibly arising from dataset size mismatches. We note that a similar objective is considered in ClipCov [17] for selecting a subset of image-text pairs for data-efficient contrastive pretraining. However, due to the combinatorial complexity of subset selection, ClipCov approximates the objective using heuristics such as maximizing CLIP similarity scores [13]. In contrast, CovMatch directly optimizes the synthetic dataset in continuous space to minimize the cross-covariance alignment loss.

**Feature Matching Loss** While aligning the cross-covariance matrices between the real and synthetic datasets is central to multimodal dataset distillation, relying solely on this can lead to suboptimal solutions. As shown in Fig. 4(a), the cross-covariance matrix $C^{\mathcal{T}}$ derived from the full Flickr30K dataset exhibits a low-rank structure, indicating that matching it alone may impose insufficient constraints, particularly as the synthetic dataset grows. Fig. 4(b) further shows that optimizing only for cross-covariance can result in notable mismatches in the mean feature embeddings.

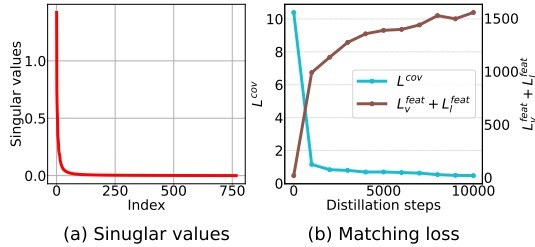

(a) Sinugular values  (b) Matching loss

Figure 4: (a) Singular values of the cross-covariance matrix $C^{\mathcal{T}}$ computed on the full Flickr30K dataset, showing its low-rank structure. (b) Evolution of the cross-covariance matching loss $\mathcal{L}^{\text{cov}}$ (11) and the feature matching loss $\mathcal{L}_v^{\text{feat}} + \mathcal{L}_l^{\text{feat}}$ (12) during distillation with only the cross-covariance objective for 500 synthetic pairs.

To mitigate this, we introduce a feature matching loss as a regularizer, which minimizes the maximum mean discrepancy between projected real and synthetic features, computed separately for each modality. For each

---
**Algorithm 1** Dataset Distillation via Cross-Covariance Matching (CovMatch)

---
1: **Require:** Full training set $\mathcal{T}$, pretrained weights $\theta_v^{\text{pretrained}}$ for image encoder $f_v$, pretrained
    weights $\theta_l^{\text{pretrained}}$ for text encoder $f_l$, the learning rate $\alpha$ for the distilled data
2: **repeat**
3:     Initialize $\theta_v, \theta_l \leftarrow \theta_v^{\text{pretrained}}, \theta_l^{\text{pretrained}}$
4:     Randomly initialize projection layers $G_v, G_l$
5:     **for** $t = 0$ to $T - 1$ **do**
6:         Sample mini-batch pairs $B^{\mathcal{T}} \sim \mathcal{T}$ and $B^{\mathcal{S}} \sim \mathcal{S}$
7:         Compute the matching loss $\mathcal{L}^{\text{CovMatch}}$ (13) on $(B^{\mathcal{T}}, B^{\mathcal{S}})$
8:         Update $\mathcal{S} \leftarrow \mathcal{S} - \alpha \cdot \nabla_{\mathcal{S}} \mathcal{L}^{\text{CovMatch}}$
9:         Train the model $(\theta_v, \theta_l, G_v, G_l)$ with $\mathcal{T}$ for one step.
10:    **end for**
11: **until** convergence
12: **Output:** $\mathcal{S}$

---

modality $m \in \{v, l\}$ (vision, language), the loss is defined as:

$$\mathcal{L}_m^{\text{feat}}(\mathcal{T}, \mathcal{S}) = \left\| \frac{1}{|\mathcal{T}|} \sum_{i=1}^{|\mathcal{T}|} G_m \cdot f_m(x_m^i; \theta_m) - \frac{1}{|S|} \sum_{j=1}^{|\mathcal{S}|} G_m \cdot f_m(\hat{x}_m^j; \theta_m) \right\|^2, \quad m \in \{v, l\}. \quad (12)$$

Here, $\{(x_v^i, x_l^i)\}_{i=1}^{|\mathcal{T}|}$ and $\{(\hat{x}_v^i, \hat{x}_l^i)\}_{i=1}^{|\mathcal{S}|}$ denote the real and synthetic datasets, respectively.

**Training Algorithm for CovMatch**    Our final objective combines cross-covariance alignment and feature matching losses:

$$\mathcal{L}^{\text{CovMatch}} = \mathcal{L}^{\text{cov}} + \lambda \cdot (\mathcal{L}_v^{\text{feat}} + \mathcal{L}_l^{\text{feat}}), \quad (13)$$

where $\lambda$ balances the contributions of the two terms. At each distillation step, the synthetic dataset $\mathcal{S}$ is updated to minimize this combined loss.

To prevent overfitting to a fixed model state, we apply an **online model update** to the image and text encoders using a batch of real data for a single gradient step before each distillation step. For additional stability, we periodically reinitialize the model: every $T$ steps, the encoders are reset to their pretrained weights and the projection layers are randomly reinitialized. The full training algorithm is presented in Algorithm 1. Figure 3 provides an overview of CovMatch, aligning cross-covariance and feature distributions between real and synthetic data.

## 4 Experimental Results

### 4.1 Experiment Setup

**Dataset and Tasks**    We evaluate the effectiveness of **CovMatch** on the Flickr30K [35] and COCO [26] datasets, following prior work [43, 45]. Both datasets consist of image-caption pairs: Flickr30K contains approximately 31K images and COCO contains 123K images, each annotated with five captions. We adopt the Karpathy split [18] for both datasets, yielding train/validation/test splits of 29K/1K/1K for Flickr30K and 113K/5K/5K for COCO, respectively. We assess model performance using a cross-modal retrieval task, where the objective is to retrieve the correct item from one modality (image or text) given a query from the other. Specifically, we compute cosine similarity between embeddings to retrieve the top-$K$ closest matches, and evaluate how often the correct match is included. We denote text-to-image retrieval as IR@K (i.e., retrieving the relevant image given a text query), and image-to-text retrieval as TR@K (i.e., retrieving the relevant text given an image query).

**Network Architectures**    Following [43, 45], we use an ImageNet-pretrained Normalizer-Free ResNet (NFNet) [4] as the image encoder and a pretrained BERT-base model [9] as the text encoder. Each encoder is followed by a trainable linear projection layer that maps features into a shared embedding space. Additional results with alternative architectures are reported in the Appendix E.1.

Table 2: Performance comparison on Flickr30K and COCO with various number of synthetic pairs. Using 100, 200, 500 pairs corresponds to approximately 0.3%, 0.7%, 1.7% of the full dataset for Flickr30k, and 0.1%, 0.2%, 0.4% for COCO. The performance achieved by training on the full dataset is as follows: IR@1=48.7, IR@5=79.2, IR@10=87.2, TR@1=61.6, TR@5=85.9, and TR@10=91.5 for Flickr30k, and IR@1=25.1, IR@5=53.9, IR@10=67.5, TR@1=33.0, TR@5=62.8, TR@10=75.0. for COCO. The reported values are averages of five runs, and the full results with standard deviations are provided in the Table 14 and Table 15.

| Pairs | Method | Flickr30k | | | | | | | COCO | | | | | | |
|---|---|---|---|---|---|---|---|---|---|---|---|---|---|---|---|
| | | IR@1 | IR@5 | IR@10 | TR@1 | TR@5 | TR@10 | Avg | IR@1 | IR@5 | IR@10 | TR@1 | TR@5 | TR@10 | Avg |
| 100 | Random | 2.0 | 7.5 | 12.6 | 3.3 | 10.4 | 16.0 | 8.6 | 0.7 | 2.8 | 5.1 | 1.0 | 4.0 | 6.9 | 3.4 |
| | Herding | 2.2 | 8.0 | 13.4 | 3.0 | 9.9 | 15.6 | 8.7 | 0.7 | 2.9 | 5.3 | 1.1 | 4.1 | 6.8 | 3.5 |
| | K-Center | 2.0 | 7.6 | 13.0 | 2.8 | 9.7 | 16.4 | 8.6 | 0.7 | 3.2 | 6.0 | 0.9 | 4.2 | 7.6 | 3.8 |
| | MTT-VL | 4.7 | 15.7 | 24.6 | 9.9 | 28.3 | 39.1 | 20.4 | 1.3 | 5.4 | 9.5 | 2.5 | 10.0 | 15.7 | 7.4 |
| | LoRS | 8.3 | 24.1 | 35.1 | 11.8 | 35.8 | 49.2 | 27.4 | 1.8 | 7.1 | 12.2 | 3.3 | 12.2 | 19.6 | 9.4 |
| | CovMatch | **10.1** | **28.6** | **40.9** | **14.8** | **38.0** | **50.6** | **30.5** | **2.8** | **10.5** | **17.7** | **3.8** | **13.1** | **21.1** | **11.5** |
| 200 | Random | 3.3 | 11.5 | 18.4 | 5.7 | 15.8 | 23.9 | 13.1 | 1.1 | 4.6 | 8.3 | 1.7 | 6.5 | 11.1 | 5.6 |
| | Herding | 3.0 | 11.3 | 18.3 | 4.7 | 15.4 | 22.9 | 12.6 | 1.2 | 4.7 | 8.5 | 1.6 | 6.6 | 11.2 | 5.6 |
| | K-Center | 3.2 | 11.1 | 17.7 | 5.3 | 15.2 | 23.2 | 12.6 | 1.2 | 5.1 | 8.9 | 1.9 | 6.7 | 11.6 | 5.9 |
| | MTT-VL | 4.6 | 16.0 | 25.5 | 10.2 | 28.7 | 41.9 | 21.2 | 1.7 | 6.5 | 12.3 | 3.3 | 11.9 | 19.4 | 9.2 |
| | LoRS | 8.6 | 25.3 | 36.6 | 14.5 | 38.7 | 53.4 | 29.5 | 2.4 | 9.3 | 15.5 | 4.3 | 14.2 | 22.6 | 11.4 |
| | CovMatch | **12.3** | **33.6** | **45.8** | **17.4** | **41.7** | **55.8** | **34.4** | **3.8** | **13.4** | **21.8** | **5.3** | **17.3** | **27.0** | **14.8** |
| 500 | Random | 6.9 | 21.0 | 31.2 | 10.0 | 28.0 | 38.7 | 22.6 | 2.2 | 8.8 | 14.9 | 3.5 | 11.9 | 19.2 | 10.1 |
| | Herding | 6.8 | 20.8 | 30.9 | 9.3 | 26.4 | 36.8 | 21.8 | 2.3 | 8.8 | 14.8 | 2.9 | 11.2 | 18.9 | 9.8 |
| | K-Center | 6.9 | 22.1 | 32.2 | 10.6 | 29.5 | 40.6 | 23.7 | 2.4 | 9.0 | 15.4 | 3.6 | 12.4 | 20.0 | 10.5 |
| | MTT-VL | 6.6 | 20.2 | 30.0 | 13.3 | 32.8 | 46.8 | 25.0 | 2.5 | 8.9 | 15.8 | 5.0 | 17.2 | 26.0 | 12.6 |
| | LoRS | 10.0 | 28.9 | 41.6 | 15.5 | 39.8 | 53.7 | 31.6 | 2.8 | 9.9 | 16.5 | 5.3 | 18.3 | 27.9 | 13.5 |
| | CovMatch | **14.7** | **38.4** | **51.4** | **19.9** | **46.7** | **59.5** | **38.4** | **5.4** | **18.0** | **28.2** | **8.1** | **23.5** | **34.6** | **19.6** |

Table 3: Cross-Architecture evaluation results on Flickr30K using 100 synthetic pairs. Reported values are averaged over six retrieval metrics: IR@1, IR@5, IR@10, TR@1, TR@5, and TR@10. Full results are provided in Table 16.

| Text encoder | BERT | | | | DistilBERT | | | |
|---|---|---|---|---|---|---|---|---|
| Image encoder | NFNet | NF-ResNet | NF-RegNet | ViT | NFNet | NF-ResNet | NF-RegNet | ViT |
| Random | 8.4 | 8.9 | 8.3 | 10.8 | 9.4 | 10.2 | 8.7 | 11.5 |
| MTT-VL | 20.4 | 8.4 | 7.5 | 9.6 | 20.2 | 7.5 | 7.0 | 8.5 |
| LoRS | 28.1 | 8.8 | 8.4 | 9.3 | 23.5 | 8.9 | 8.3 | 8.9 |
| CovMatch | **30.2** | **15.5** | **14.6** | **15.1** | **27.1** | **16.1** | **14.6** | **13.4** |

**Baselines** We compare CovMatch against both coreset selection and dataset distillation methods. For coreset selection, we include Random (uniform sampling), Herding [7], and K-Center [38], all adapted to the multimodal setting as in [43]. For dataset distillation, we evaluate MTT-VL [43] and LoRS [45]. Following their original protocols, these methods freeze the text encoder during both distillation and evaluation due to computational and optimization constraints. In contrast, all other baselines fine-tune the entire network. Further implementation details are available in Appendix B.

## 4.2 Main Results

**Flickr30K and COCO** We evaluate CovMatch on both Flickr30K and COCO, comparing its performance against several baselines across varying numbers of synthetic image-text pairs, as presented in Table 2. A key observation is that performance gains from existing dataset distillation methods quickly saturate as the number of synthetic pairs increases—even in the extremely low-data regime (i.e., under 2% of the full dataset). This result indicates that fine-tuning the text encoder is critical for effective multimodal contrastive learning, particularly when scaling to larger synthetic datasets. By jointly optimizing both the image and text encoders through cross-covariance alignment, CovMatch consistently outperforms prior methods and establishes new state-of-the-art results across all settings. Notably, on Flickr30K with 500 synthetic pairs, CovMatch achieves a 6.8% absolute improvement over the strongest baseline.

**Cross-Architecture Generalization** One of the essential characteristics of a distilled dataset is its ability to generalize across different, unseen architectures. To demonstrate CovMatch's cross-architecture generalizability, we distill the dataset with NFNet and BERT, and then evaluate it with alternative architectures—NF-ResNet [3], NF-RegNet [44], ViT [10] for the image encoder, and DistilBERT [37] for the text encoder. As shown in Table 3, the dataset distilled by CovMatch

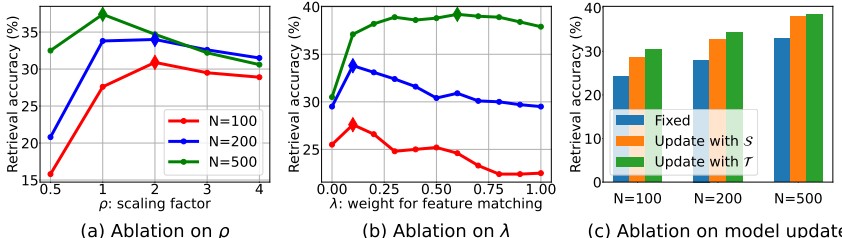

(a) Ablation on $\rho$     (b) Ablation on $\lambda$     (c) Ablation on model update

Figure 5: (a) Ablation on the scaling factor $\rho$ in (11). Scaling the real cross-covariance $C^{\mathcal{T}}$ is particularly important when $N$ is small. (b) Ablation on the feature matching weight $\lambda$ in (13). The optimal value of $\lambda$ tends to increase as $N$ increases. (c) Ablation on the online model update strategy. Updating with real data exhibits the best performance. All ablation studies are conducted on Flickr30k with varying $N$.

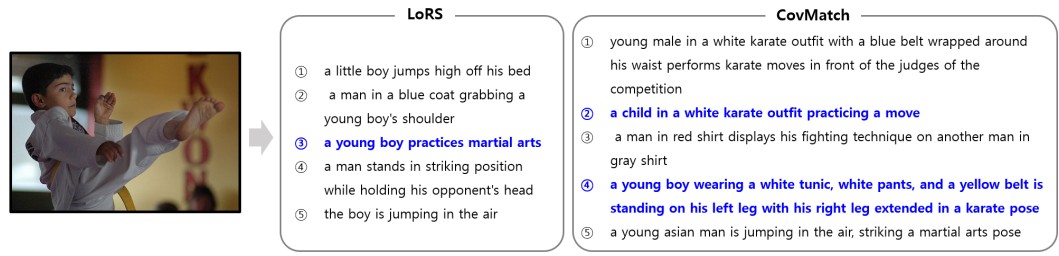

Figure 6: Comparison of the top five retrieved captions from models trained with LoRS- and CovMatch-generated synthetic image-text pairs, given an image query. Each model is trained on 500 synthetic pairs, and queries are from the Flickr30K test set. Ground-truth captions (i.e., five captions paired with the query image in Flickr30K test set) are highlighted in blue.

generalizes effectively to both unseen image and text encoders, whereas baseline methods perform comparably to or worse than random selection. We attribute this to the fact that prior methods only optimize the image encoder and projection layers during distillation, which induces overfitting to the specific architecture used in training.

### 4.3 Ablation Study and Further Analysis

**Scaling Factor $\rho$**    CovMatch introduces a scaling factor $\rho$ in (11) when aligning the cross-covariance matrix of the synthetic dataset with that of the real dataset, where $\rho$ originates from the regularization term in the inner optimization of the original bi-level optimization problem (9). Figure 5(a) presents the impact of varying $\rho$ across different numbers of synthetic image-text pairs. The results indicate that appropriate scaling becomes particularly important when the number of synthetic samples is small. This finding is consistent with the intuition that stronger regularization is required in low-data regimes to prevent overfitting.

**Feature Matching**    We also investigate the effect of the feature matching objective by varying its weighting coefficient $\lambda$ in (13). As shown in Figure 5(b), incorporating feature matching significantly enhances performance, particularly when a larger number of synthetic pairs is used. Moreover, the optimal value of $\lambda$ tends to increase with the number of pairs. This observation is consistent with our intuition: as the number of synthetic samples grows, aligning cross-covariance—a second-order statistic—becomes easier, even when the underlying feature distributions remain misaligned. Consequently, stronger feature-level regularization is required to constrain the optimization toward semantically meaningful representations.

**Online Model Update**    Figure 5(c) shows the impact of the online model update. We compare three update strategies: (1) fixing the encoder, (2) updating the encoder with the synthetic set $\mathcal{S}$, and (3) updating it with the real data from $\mathcal{T}$—the latter being the strategy adopted in our method. The result reveals that updating the online model is crucial for preventing overfitting to a fixed model

state. Additionally, further improvements are achieved by updating the online model with real data from $\mathcal{T}$, rather than with the synthetic dataset $\mathcal{S}$.

**Qualitative Analysis** In contrast to previous methods, CovMatch incorporates the text encoder into both the distillation and evaluation processes. To illustrate the impact of this design, we provide text-retrieval examples in Figure 6. For each image query, we show the top five retrieved captions using models trained on synthetic image-text pairs generated by LoRS and CovMatch, respectively; the ground-truth caption (from the five true captions paired with the image) is highlighted in blue. As shown, CovMatch leads to stronger alignment between visual and textual modalities, retrieving more ground-truth captions. While LoRS often captures only basic semantics (e.g., "boy"), CovMatch enables alignment with more nuanced concepts (e.g., "karate"), reflecting the tighter clustering of text representations seen in Figure 2(a). More examples are provided in Appendix F.

# 5 Conclusion

In this work, we revisit the original bi-level optimization formulation of dataset distillation and, under the assumption of linear contrastive learning, derive a simplified objective that facilitates the inclusion of the text encoder in the multimodal dataset distillation framework. We present CovMatch, a lightweight and scalable algorithm for multimodal dataset distillation that aligns the cross-covariance of image-text embeddings between real and synthetic datasets, with additional regularization via feature distribution alignment within each modality. CovMatch outperforms existing algorithms with significant performance improvements, enhancing cross-modal retrieval ability, cross-architecture generalization and scalability.

**Limitations and Future Works** We assume a scenario in which image and text encoders, pre-trained within their respective modalities, are available. Accordingly, our method focuses on dataset distillation to effectively fine-tune image-text contrastive models. However, we have not yet explored scenarios where pretrained image and text encoders are unavailable, requiring training from scratch. Distilling datasets for pretraining image-text contrastive models presents a promising direction for future work.

**Broader Impact** This paper introduces a computationally and memory-efficient multimodal dataset distillation method. CovMatch demonstrates strong generalization across different network architectures, highlighting its versatility and robustness. These features, combined with its enhanced scalability, make it a highly practical solution for real-world applications. Additionally, our new framework, which freezes only the input embedding layer while incorporating the transformer layers of the text encoder in the distillation process, provides a solid foundation for future research and serves as a starting point for the development of more efficient and scalable multimodal learning methods.

## Acknowledgments and Disclosure of Funding

This work was supported by the National Research Foundation of Korea (NRF) grant funded by the Korea government (MSIT) (No. RS-2024-00408003, RS-2025-00516153 and RS-2024-00444862).

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

# A   Related Works

**Traditional Dataset Distillation**   Dataset distillation, introduced in [41], aims to create a small synthetic set $\mathcal{S}$, so that a model $\theta^{\mathcal{S}}$ trained on $\mathcal{S}$ achieves good generalization performance, performing well on the full dataset $\mathcal{T}$:

$$\mathcal{S}^* = \arg\min_{\mathcal{S}} \mathcal{L}^{\mathcal{T}}(\theta^{\mathcal{S}}), \text{ with } \theta^{\mathcal{S}} = \arg\min_{\theta} \mathcal{L}^{\mathcal{S}}(\theta),$$

Here, $\mathcal{L}^{\mathcal{T}}$ and $\mathcal{L}^{\mathcal{S}}$ are losses on $\mathcal{T}$ and $\mathcal{S}$, respectively. To address the bi-level optimization's computational complexity and memory demands, existing works have employed two methods: closed-form approximation and surrogate-based matching.

Closed-form approximation methods obtain meta-gradient $\nabla_S \mathcal{L}^{\mathcal{T}}(\theta^{\mathcal{S}})$ in closed-form by approximating the inner optimization. For example, KIP [32, 33] and FrePo [54] obtain closed-form solution for inner optimization by approximating it to kernel ridge regression. RCIG [27] assumes that the inner optimization is convex and computes the meta-gradient via implicit gradient.

Surrogate-based matching methods replace the original objective with a simpler surrogate objective. It can be further categorized into two groups: short-term matching and long-term matching. Short-term matching methods aim to optimize the synthetic dataset such that it emulates the short-range behavior of the model with a given parameter $\theta$. For example, DC [53], DSA [51], and DCC [21] matches gradient of neural network computed with synthetic and real dataset $\nabla_\theta \mathcal{L}^{\mathcal{S}}(\theta)$ and $\nabla_\theta \mathcal{L}^{\mathcal{T}}(\theta)$. DM [52] and CAFE [40] matches distribution of $\mathcal{S}$ and $\mathcal{T}$ in the feature space. Since these approaches do not involve unrolled optimization, they are computationally efficient. In contrast, long-term matching methods, such as MTT [5] and TESLA [8], aim to emulate long-term behavior by matching network parameters after several steps of training on the synthetic dataset. Although long-term matching has demonstrated superior performance compared to short-term matching methods, it introduces substantial costs. While TESLA [8] improves memory efficiency by making memory usage constant with respect to the number of synthetic steps, it still inherits the drawbacks of trajectory preparation and high computational overhead inherent to long-term matching.

**Image-Text Retrieval**   Image-text retrieval, the canonical multimodal retrieval task, aims to align visual and textual representations within a shared semantic space. Early methods such as De-ViSE [12] and Deep CCA [1] introduce joint embedding frameworks, later enhanced by VSE++ [11] through hard-negative mining. Fine-grained cross-encoder models like SCAN [20] and VSRN [24] capture region–word interactions to improve alignment accuracy, though at the expense of efficiency. The paradigm has shifted with large-scale contrastive dual-encoders—notably CLIP [36] and ALIGN [16]—which trained image–text encoders on web-scale noisy data, establishing a foundation for efficient zero-shot retrieval. Building on this, LiT [50] enhances data efficiency by freezing the image encoder, SLIP [30] integrates self-supervised objectives, and SigLIP [49] introduces a pairwise sigmoid loss for improved stability. To retain fine-grained matching without full cross-attention, FILIP [46] employs late interaction between token–patch pairs, while fusion-based pretraining frameworks such as ALBEF [23], BLIP [22], and CoCa [47] combine contrastive learning with image-conditioned language modeling to mitigate noisy supervision. Beyond empirical successes, a few recent works have provided theoretical insights into multimodal contrastive learning. The work in [31] showes that for linear models, each step of loss minimization via gradient descent can be interpreted as performing singular value decomposition (SVD) on the contrastive cross-covariance matrix between modalities, which forms a key insight motivating our work.

**Multimodal Dataset Distillation**   Recent efforts have extended dataset distillation techniques to the image-text contrastive learning setting. MTT-VL [43] adapts the MTT [5] framework in a naive manner, matching the parameters of the image and text encoders independently. LoRS [45] further improves performance by jointly distilling the ground-truth similarity matrix between image-text pairs, effectively capturing cross-modal correspondence. In addition to contrastive tasks, other works have explored dataset distillation in different domains and problem settings. The work in [19] addresses a multimodal classification task with audio and video modalities by matching cross-modal distributions, while [25, 28] investigate dataset distillation for natural language processing tasks and [42, 6] extend it to video classification. In parallel, ClipCov [17] proposes a dataset selection method for multimodal contrastive learning, aiming to preserve the cross-covariance structure of the original dataset. While ClipCov relies on heuristic surrogate objectives to approximate cross-covariance preservation, our work, CovMatch, directly optimizes the synthetic dataset to achieve this objective.

# B   Implementation Details

**Distillation**   The synthetic dataset is initialized with randomly selected real image-text pairs from the training set. During distillation, both the synthetic image pixels and text input embeddings are optimized using SGD with momentum 0.5 and a learning rate of 1.0. At each distillation step, the cross-covariance matrix and feature means are computed using a batch of 128 real samples. For the synthetic data, the entire set is used for these computations, except in the 500-pair setting, where a batch of 256 synthetic samples is used to reduce memory consumption. We set the scaling factor to $\rho = 2$ for 100 synthetic pairs and $\rho = 1$ for 200 or more pairs. The feature matching weight $\lambda$ is fixed at 0.1 for 100 and 200 pairs, and increased to 0.5 or 0.6 for 500 pairs to impose stronger regularization on cross-covariance alignment. Note that all network components—including the image encoder, text encoder, and projection layers—are updated with one step of training on the real dataset at each distillation step, and re-initialized every 50 updates. We distill for 10,000 iterations by default; for the 500-pair setting, we extend this to 20,000 iterations to ensure full convergence, even after reaching 95% of the final performance. A summary of the hyperparameter used in CovMatch is provided in Table 4.

Table 4: Hyper-parameters used for CovMatch. A dash (-) in the synthetic batch size denotes that the entire synthetic set is used for computing the matching loss.

| Dataset | Flickr30k | | | COCO | | |
|---|---|---|---|---|---|---|
| Pairs | 100 | 200 | 500 | 100 | 200 | 500 |
| Scaling factor $\rho$ | 2 | 1 | 1 | 2 | 1 | 1 |
| Feature match weight $\lambda$ | 0.1 | 0.1 | 0.6 | 0.1 | 0.1 | 0.5 |
| Batch size (real) | 128 | 128 | 128 | 128 | 128 | 128 |
| Batch size (syn) | - | - | 256 | - | - | 256 |
| Learning rate (img) | 1 | 1 | 1 | 1 | 1 | 1 |
| Learning rate (txt) | 1 | 1 | 1 | 1 | 1 | 1 |
| Iteration | 200 | 200 | 400 | 200 | 200 | 400 |

**Evaluation**   During the evaluation stage, we train the model using SGD optimizer with momentum 0.9, weight decay 5e-4, batch size 128, and learning rate 0.01 for the image and text encoders and 0.1 for the projection layers. Training is conducted for 100 epochs, and we employ a multi-step learning rate scheduler that decays the learning rate by a factor of 0.1 at the 50th epoch.

**Network Architectures**   In Table 5, we provide detailed information on all the models used in our experiments. For image encoders, we use NFNet, NF-ResNet, NF-RegNet, and ViT, and for text encoders, we employ BERT and DistilBERT.

Table 5: Detailed information of the models.

| Network Architecture | Model | Parameter Count |
|---|---|---|
| NFNet [4] | nfnet_l0 | 26.4M |
| NF-ResNet [3] | nf_resnet50 | 25.6M |
| NF-RegNet [44] | nf_regnet_b1 | 21.0M |
| ViT [10] | vit_base_patch16_224 | 85.8M |
| BERT [9] | bert-base-uncased | 110.0M |
| DistilBERT [37] | distilbert-base-uncased | 66.0M |

# C   Discussion about Linearized Contrastive Learning

In Section 3.1, we derived the cross-covariance alignment objective under the linearized contrastive learning assumption. In this section, we provide theoretical justification for using the linear contrastive loss (5) and full derivation for Equation (6).

**Theoretical Justification for Linearized Contrastive Objective** We note that the linear contrastive loss (5) can be interpreted as a high-temperature approximation of the InfoNCE loss (1). Specifically, under the assumption of a large temperature $\tau > 0$, the softmax in the InfoNCE loss flattens, and a first-order Taylor expansion yields

$$\log(\sum_{j \neq i} \exp(s_{ij}/\tau)) \approx \log \Big( \sum_{j \neq i} \Big( 1 + \frac{s_{ij}}{\tau} \Big) \Big)$$

$$= \log \Big( \frac{1}{M-1} \sum_{j \neq i} \Big( 1 + \frac{s_{ij}}{\tau} \Big) \Big) + \log(M-1)$$

$$\approx \frac{1}{\tau(M-1)} \sum_{j \neq i} s_{ij} + \log(M-1).$$

Substituting this into the InfoNCE loss (1) leads to the linearized form

$$\mathcal{L}_{\text{NCE}} \approx \frac{1}{\tau M(M-1)} \sum_{i=1}^{M} \sum_{j \neq i} (s_{ij} - s_{ii}) + \frac{1}{\tau M(M-1)} \sum_{i=1}^{M} \sum_{j \neq i} (s_{ji} - s_{ii}) + \text{Constant},$$

which is equivalent to (5), up to a constant scaling and shift. This derivation provides justification for using the linearized objective as an approximation of InfoNCE when $\tau$ is large.

Also, recent work has shown that fine-tuning large neural networks (e.g., foundation models) often operates in the Neural Tangent Kernel (NTK) regime [14], where the training dynamics are well approximated by a linear model over a high-dimensional feature space derived from the model's gradients [29]. This connection suggests that our analysis in the linear regime can meaningfully inform behavior in more general, non-linear settings.

**Equivalence between Eq. (5) and Eq. (6)** Remind that the cross-covariance matrix of the dataset $\mathcal{D} = \{(h_v^i, h_l^i)\}_{i=1}^{|\mathcal{D}|}$ is defined as

$$C^{\mathcal{D}} = \frac{1}{(|\mathcal{D}|-1)} \sum_{i=1}^{|\mathcal{D}|} (h_v^i - \mu_{h_v})(h_l^i - \mu_{h_l})^{\top} = \frac{1}{(|\mathcal{D}|-1)} (\sum_{i=1}^{|\mathcal{D}|} h_v^i (h_l^i)^{\top} - |\mathcal{D}| \mu_{h_v}(\mu_{h_l})^{\top}), \quad (14)$$

with $\mu_{h_v}$ and $\mu_{h_l}$ denoting the empirical means of the image and text features, respectively. The cross-covariance alignment term can be written as

$$- \operatorname{Tr}(G_v C^{\mathcal{D}} G_l^{\top}) = - \sum_{k=1}^{z} g_{v_k}^{\top} C^{\mathcal{D}} g_{l_k}, \quad (15)$$

where $G_v^{\top} = [g_{v_1}, \ldots, g_{v_z}] \in \mathbb{R}^{d_v \times z}$ and $G_l^{\top} = [g_{l_1}, \ldots, g_{l_z}] \in \mathbb{R}^{d_l \times z}$. Note that the similarity terms in (5) can be expressed as

$$s_{ij} := (G_v h_v^i)^{\top} (G_l h_l^j) = \sum_{k=1}^{z} (g_{v_k})^{\top} h_v^i (h_l^j)^{\top} g_{l_k}.$$

Then, the linear contrastive loss (5) can be written as:

$$\mathcal{L}_{\text{lin}}(G_v, G_l; \mathcal{D}) = \frac{1}{|\mathcal{D}|(|\mathcal{D}|-1)} \sum_{i=1}^{|\mathcal{D}|} \sum_{j=1}^{|\mathcal{D}|} s_{ij} - \frac{1}{|\mathcal{D}|-1} \sum_{i=1}^{|\mathcal{D}|} s_{ii} + \frac{\rho}{2} \|G_v^{\top} G_l\|_F^2$$

$$= \sum_{k=1}^{z} g_{v_k}^{\top} \Bigg[ \frac{1}{|\mathcal{D}|(|\mathcal{D}|-1)} \sum_{i=1}^{|\mathcal{D}|} \sum_{j=1}^{|\mathcal{D}|} h_v^i (h_l^j)^{\top} - \frac{1}{|\mathcal{D}|-1} \sum_{i=1}^{|\mathcal{D}|} h_v^i (h_l^i)^{\top} \Bigg] g_{l_k} + \frac{\rho}{2} \|G_v^{\top} G_l\|_F^2$$

$$= \sum_{k=1}^{z} g_{v_k}^{\top} \Bigg[ \frac{|\mathcal{D}|}{(|\mathcal{D}|-1)} \mu_{h_v}(\mu_{h_l})^{\top} - \frac{1}{|\mathcal{D}|-1} \sum_{i=1}^{|\mathcal{D}|} h_v^i (h_l^i)^{\top} \Bigg] g_{l_k} + \frac{\rho}{2} \|G_v^{\top} G_l\|_F^2$$

$$= - \operatorname{Tr}(G_v C^{\mathcal{D}} G_l^{\top}) + \frac{\rho}{2} \|G_v^{\top} G_l\|_F^2.$$

Hence, the linear contrastive loss (5) is equivalent to cross-covariance alignment term with a regularization term on the projection matrices (6).

# D Generalization to Video-Text Retrieval Tasks

To further evaluate the generalizability of CovMatch beyond image–text data, we conduct experiments on the video–text retrieval task. For this study, we construct a computationally-manageable subset of the WebVid-10M dataset [2], consisting of 49K training and 1K test video–text pairs. We use a pretrained VideoResNet (r3d_18) [39] as the video encoder and BERT as the text encoder. For the synthetic dataset, we distill 500 video–text pairs, sampling four frames per video.

As shown in Table 6, CovMatch outperforms both MTT-VL [43] and LoRS [45], demonstrating superior retrieval performance. The result indicates that CovMatch generalizes effectively to the video-text retrieval task, underscoring its robustness across modalities. Its strong performance on this more complex task also highlights the CovMatch's computational efficiency.

Table 6: Performance comparison on video-text retrieval task with $N = 500$. The performance achieved by training on the full dataset is as follows: VR@1=15.2, VR@5=38.9, VR@10=53.6, TR@1=14.2, TR@5=38.4, TR@10=52.8.

| Method | VR@1 | VR@5 | VR@10 | TR@1 | TR@5 | TR@10 | Avg |
|--------|------|------|-------|------|------|-------|-----|
| Random | 1.1 | 5.2 | 9.6 | 1.8 | 5.8 | 10.2 | 5.6 |
| MTT-VL | 2.0 | 8.3 | 13.9 | 2.2 | 8.8 | 13.7 | 8.1 |
| LoRS | 2.4 | 8.7 | 14.1 | 2.8 | 8.5 | 13.4 | 8.3 |
| CovMatch | **3.0** | **11.4** | **19.0** | **3.4** | **11.1** | **18.0** | **11.0** |

# E More Ablation Studies

## E.1 Distillation with Other Networks

We use NFNet as the image encoder and BERT as the text encoder for our main results. In this section, we evaluate the generalization ability of our method by applying it to alternative network architectures for both the image and text encoders. The results, summarized in Table 7, demonstrate that CovMatch consistently outperforms all baseline methods across a range of encoder choices, confirming its robustness to architectural variations.

Table 7: Distillation with various network architectures for both image and text encoders on Flickr30k using 100 synthetic pairs. The average retrieval performance is reported. Note that for MTT-VL with the ViT model, the low-rank adaptation matching technique is employed, which involves matching the trajectories of a small subset of model parameters using low-rank matrices.

| Image encoder | Text encoder | Random | MTT-VL | LoRS | CovMatch |
|---------------|--------------|--------|--------|------|----------|
| NFNet | BERT | 8.6 | 20.4 | 27.4 | **30.5** |
| NF-ResNet | BERT | 9.1 | 14.4 | 20.7 | **25.3** |
| NF-RegNet | BERT | 7.7 | 16.6 | 22.0 | **26.6** |
| ViT | BERT | 11.3 | 20.7 | 29.9 | **35.9** |
| NFNet | DistilBERT | 9.5 | 24.0 | 26.0 | **29.1** |

## E.2 Ablation on Frozeness of Text Encoder

As shown in Figure 1, our method distills the text modality into synthetic embeddings used as inputs to the transformer layers of the text encoder, whereas baseline methods distill text embeddings as inputs to the text projection head, bypassing the text encoder entirely. Consequently, under the baselines, the text encoder is never involved—even at evaluation. To enable a fairer comparison in which the text encoder can be trained during evaluation for the baselines, we adapt our framework (Figure 1) to them: instead of feeding distilled text embeddings directly into the projection head, we modify the baselines to distill synthetic embeddings that serve as inputs to the text encoder's transformer layers. In this setup, we freeze the text encoder during distillation and unfreeze it only at

evaluation. We made this choice for two reasons: (i) unfreezing the text encoder during distillation incurs substantial compute and storage overhead; and (ii) despite the high cost, when we did unfreeze it and included it in trajectory matching, the text-encoder trajectory-matching loss did not decrease (unlike the projection head), leading to optimization failure and unstable training.

As reported in Table 8, unfreezing the text encoder only for evaluation yields a small improvement for the baselines, but our method still significantly outperforms them. This observation confirms that baseline methods, which do not incorporate the text encoder during distillation, cannot fully benefit from unfreezing it during evaluation, resulting in only marginal performance gains.

Also, we conduct an additional ablation study that investigates the effect of freezing the text encoder for our method during both the distillation and evaluation stages. As shown in Table 9, freezing the text encoder during evaluation leads to a significant performance drop, highlighting the importance of training not only the image encoder and projection layers but also the text encoder for strong image-text retrieval ability. Moreover, we observe that freezing the text encoder during distillation also causes a substantial drop in performance (from 38.4 to 29.4 in average score). These findings suggest that involving the text encoder during distillation is critical for maximizing the effectiveness of our method.

Table 8: Results of unfreezing text encoder during the evaluation stage of baseline method on Flickr30k with $N = 500$.

| Method | IR@1 | IR@5 | IR@10 | TR@1 | TR@5 | TR@10 | Avg |
|---|---|---|---|---|---|---|---|
| LoRS (frozen) | 10.0 | 28.9 | 41.6 | 15.5 | 39.8 | 53.7 | 31.6 |
| LoRS (unfrozen) | 10.0 | 28.0 | 41.5 | 17.7 | 42.5 | 57.0 | 32.8 |
| CovMatch | 14.7 | 38.4 | 51.4 | 19.9 | 46.7 | 59.5 | 38.4 |

Table 9: Effect of freezing the text encoder during the distillation and evaluation stages of CovMatch on Flickr30k with $N = 500$.

| Distill | Eval | IR@1 | IR@5 | IR@10 | TR@1 | TR@5 | TR@10 | Avg |
|---|---|---|---|---|---|---|---|---|
| Frozen | Frozen | 4.3 | 13.4 | 21.1 | 6.5 | 20.6 | 30.6 | 16.1 |
| Frozen | Unforzen | 9.8 | 27.8 | 40.2 | 15.1 | 35.8 | 48.0 | 29.4 |
| Unfrozen | Frozen | 4.5 | 14.7 | 22.6 | 8.4 | 23.2 | 33.4 | 17.8 |
| Unfrozen | Unfrozen | 14.7 | 38.4 | 51.4 | 19.9 | 46.7 | 59.5 | 38.4 |

### E.3 Ablation on Batch Size

While our original objective aims to match the cross-covariance matrix and mean feature of the entire training dataset with those of the synthetic set, computing the exact statistics over the full training set at every distillation step is computationally prohibitive, since the embeddings are continuously updated by the online model. To address this, we approximate the full-dataset statistics using mini-batches sampled from the training set at each step. This strategy introduces some variance, but allows the method to remain tractable. We investigate the impact of the size of this mini-batches in Table 10. The result shows a clear trend: larger batch sizes consistently yield better retrieval performance. This improvement arises because larger batches produce more accurate estimates of the true cross-covariance, enabling stronger alignment between real and synthetic features and thus more effective distillation. However, this performance gain comes at the cost of increased per-iteration computation time, highlighting an inherent trade-off between statistical fidelity and computational efficiency in our cross-covariance matching framework.

### E.4 Ablation on Projection Layers

We adopt a two-layer projection network with GELU activation, following prior baseline designs. To explore the design space of projection layers, we investigate two key architectural factors: projection dimension (i.e., dimension of projected output) and projection depth (i.e., number of layers in the

Table 10: Effect of batch size on performance and distillation time on Flickr30k with $N = 100$. Our default configuration uses a batch size of 128.

| Batch size | 64 | 128 | 256 | 512 | 1024 |
|---|---|---|---|---|---|
| Performance | 29.1 | 30.5 | 30.4 | 30.9 | 31.1 |
| Time (Sec/iter) | 1.02 | 1.22 | 1.36 | 1.79 | 3.50 |

Table 11: Effect of projection dimension on Flickr30k with $N$=100.

| Projection Dim | 256 | 768 | 2304 | 4096 |
|---|---|---|---|---|
| Performance | 16.1 | 26.0 | 30.5 | 28.9 |

Table 12: Effect of projection depth on Flickr30k with $N$=100.

| Projection Depth | 1 | 2 | 3 |
|---|---|---|---|
| Performance | 29.5 | 30.5 | 19.9 |

projection network). Note that our default configuration uses a projection dimension of 2304. As shown in Table 11, reducing the projection dimension leads to noticeable performance degradation, likely due to loss of information. Also, as shown in Table 12, single-layer projection network achieved performance comparable to our default two-layer setup, but deeper networks (3 layers) significantly degraded the performance. These results highlight that both the dimensionality and depth of the projection head play an important role, and an overly compact or overly deep design can hinder retrieval effectiveness.

### E.5 Ablation with Single Modality

We conduct an ablation study to assess the impact of distilling each modality individually. Specifically, we perform distillation while freezing either the synthetic image or text inputs. As shown in Table 13, freezing either modality leads to a substantial drop in performance, indicating that both image and text play essential roles in effective model training. Notably, the performance degradation is more severe when the images are frozen, suggesting that learning the image modality is more critical.

Table 13: Ablation study on single-modality distillation using 100 image-text pairs on Flickr30k. The results show that jointly optimizing both modalities is crucial for effective performance.

| Method | IR@1 | IR@5 | IR@10 | TR@1 | TR@5 | TR@10 | Avg |
|---|---|---|---|---|---|---|---|
| Image-only | 7.1 | 22.0 | 32.9 | 11.0 | 30.7 | 43.1 | 24.5 |
| Text-only | 5.8 | 20.0 | 30.6 | 8.2 | 25.1 | 35.3 | 20.8 |
| Both | 10.1 | 28.6 | 40.9 | 14.8 | 38.0 | 50.6 | 30.5 |

## F   More Qualitative Results

In Figure 7, we provide more examples of qualitative analysis described in Section 4.3.

## G   Visualizations

We present visualizations of $N = 100$ distilled image-text pairs from the COCO dataset in Figure 8. The image-text pair on the left represents the initial pair before distillation, while the image-text pair on the right corresponds to the distilled pair after distillation. We observe that the images are transformed into a DeepDream-like style [48], exhibiting high-frequency components. The text representations are visualized by retrieving the nearest caption from the training set in the text embedding space.

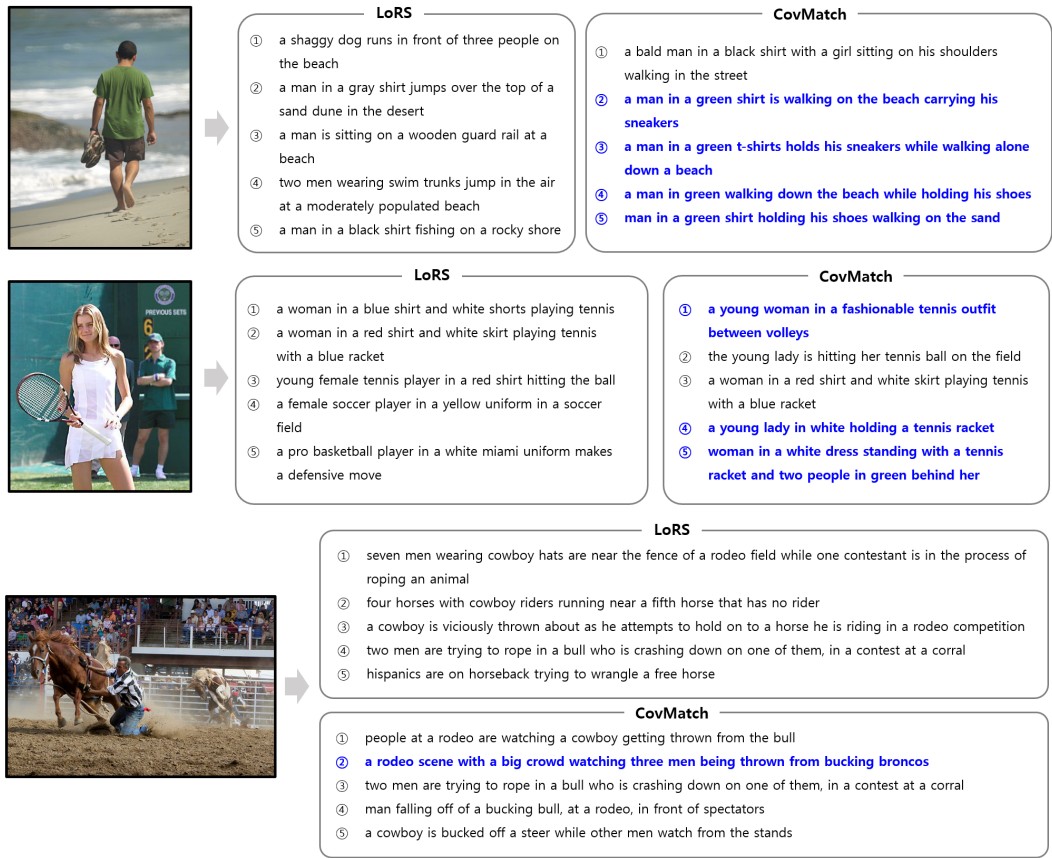

Figure 7: Comparison of the top five retrieved captions from models trained with LoRS- and CovMatch-generated synthetic image-text pairs, given an image query. Each model is trained on 500 synthetic pairs, and queries are from the Flickr30K test set. Ground-truth captions (i.e., five captions paired with the query image in Flickr30K test set) are highlighted in blue.

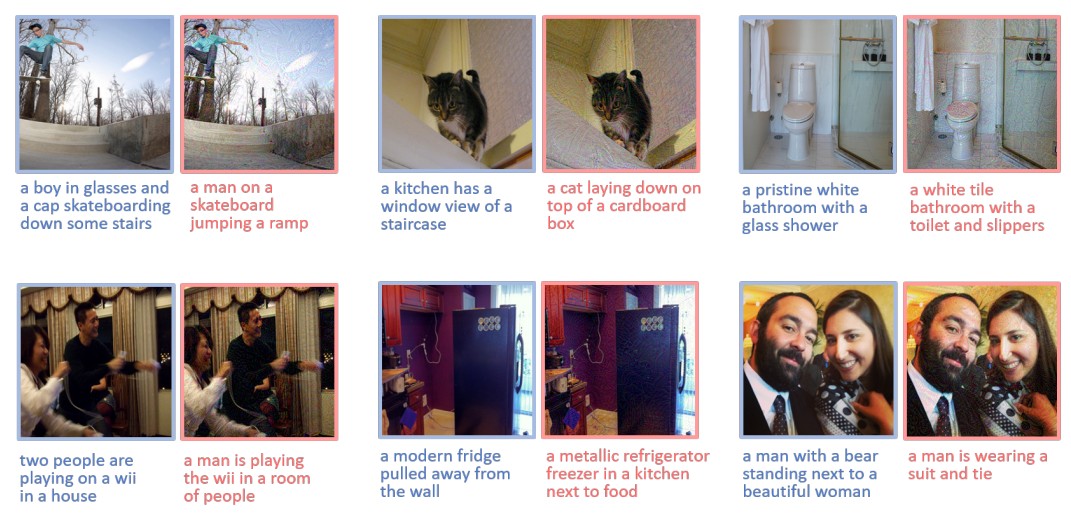

Figure 8: Examples of distilled image-text pairs from the COCO dataset with $N = 100$ pairs. (Left) Initial image-text pair before distillation. (Right) Distilled image-text pairs.

# H Full Results

Table 14 and Table 15 present the full results for Flickr30k and COCO, respectively. Table 16 shows the results of the cross-architecture generalization experiment on Flickr30k using $N = 100$ synthetic pairs.

Table 14: Performance comparison on Flickr30K. The performance trained by full dataset is IR@1=48.7, IR@5=79.2, IR@10=87.2, TR@1=61.6, TR@5=85.9, and TR@10=91.5%.

| Pairs | Ratio | Metrics | Coreset Selection | | | Distillation | | |
|-------|-------|---------|--------|---------|----------|--------|------|---------|
| | | | Random | Herding | K-Center | MTT-VL | LoRS | CovMatch |
| 100 | 0.3 % | IR@1 | 2.0±0.2 | 2.2±0.3 | 2.0±0.2 | 4.7±0.2 | 8.3±0.2 | **10.1±0.2** |
| | | IR@5 | 7.5±0.7 | 8.0±0.8 | 7.6±0.7 | 15.7±0.5 | 24.1±0.2 | **28.6±0.4** |
| | | IR@10 | 12.6±1.0 | 13.4±1.0 | 13.0±1.0 | 24.6±1.0 | 35.1±0.3 | **40.9±0.6** |
| | | TR@1 | 3.3±0.1 | 3.0±0.7 | 2.8±0.5 | 9.9±0.3 | 11.8±0.2 | **14.8±0.9** |
| | | TR@5 | 10.4±0.5 | 9.9±1.0 | 9.7±1.0 | 28.3±0.5 | 35.8±0.6 | **38.0±0.4** |
| | | TR@10 | 16.0±0.6 | 15.6±1.1 | 16.4±0.9 | 39.1±0.7 | 49.2±0.5 | **50.6±0.6** |
| 200 | 0.7 % | IR@1 | 3.3±0.2 | 3.0±0.2 | 3.2±0.1 | 4.6±0.9 | 8.6±0.3 | **12.3±0.4** |
| | | IR@5 | 11.5±0.4 | 11.3±0.4 | 11.1±0.5 | 16.0±1.6 | 25.3±0.2 | **33.6±0.3** |
| | | IR@10 | 18.4±0.5 | 18.3±0.7 | 17.7±0.4 | 25.5±2.6 | 36.6±0.3 | **45.8±0.2** |
| | | TR@1 | 5.7±0.5 | 4.7±0.4 | 5.3±0.7 | 10.2±0.8 | 14.5±0.5 | **17.4±0.5** |
| | | TR@5 | 15.8±0.5 | 15.4±0.4 | 15.2±0.9 | 28.7±1.0 | 38.7±0.5 | **41.7±0.5** |
| | | TR@10 | 23.9±1.3 | 22.9±1.0 | 23.2±0.4 | 41.9±1.9 | 53.4±0.5 | **55.8±0.5** |
| 500 | 1.7 % | IR@1 | 6.9±0.4 | 6.8±0.4 | 6.9±0.2 | 6.6±0.3 | 10.0±0.2 | **14.7±0.3** |
| | | IR@5 | 21.0±0.4 | 20.8±0.5 | 22.1±0.4 | 20.2±1.2 | 28.9±0.7 | **38.4±0.4** |
| | | IR@10 | 31.2±0.6 | 30.9±0.6 | 32.2±0.6 | 30.0±2.1 | 41.6±0.6 | **51.4±0.3** |
| | | TR@1 | 10.0±0.6 | 9.3±0.6 | 10.6±0.7 | 13.3±0.6 | 15.5±0.7 | **19.9±0.6** |
| | | TR@5 | 28.0±0.8 | 26.4±0.5 | 29.5±0.7 | 32.8±1.8 | 39.8±0.4 | **46.7±0.9** |
| | | TR@10 | 38.7±0.9 | 36.8±0.7 | 40.6±0.3 | 46.8±0.8 | 53.7±0.3 | **59.5±0.7** |

Table 15: Performance comparison on COCO. The performance trained by full dataset is IR@1=25.1, IR@5=53.9, IR@10=67.5, TR@1=33.0, TR@5=62.8, TR@10=75.0.

| Pairs | Ratio | Metrics | Coreset Selection | | | Distillation | | |
|-------|-------|---------|--------|---------|----------|--------|------|---------|
| | | | Random | Herding | K-Center | MTT-VL | LoRS | CovMatch |
| 100 | 0.1 % | IR@1 | 0.7±0.1 | 0.7±0.1 | 0.7±0.1 | 1.3±0.1 | 1.8±0.1 | **2.8±0.1** |
| | | IR@5 | 2.8±0.1 | 2.9±0.1 | 3.2±0.1 | 5.4±0.3 | 7.1±0.2 | **10.5±0.2** |
| | | IR@10 | 5.1±0.3 | 5.3±0.2 | 6.0±0.2 | 9.5±0.5 | 12.2±0.2 | **17.7±0.3** |
| | | TR@1 | 1.0±0.1 | 1.1±0.1 | 0.9±0.1 | 2.5±0.3 | 3.3±0.2 | **3.8±0.1** |
| | | TR@5 | 4.0±0.2 | 4.1±0.2 | 4.2±0.2 | 10.0±0.5 | 12.2±0.3 | **13.1±0.3** |
| | | TR@10 | 6.9±0.3 | 6.8±0.2 | 7.6±0.3 | 15.7±0.4 | 19.6±0.3 | **21.1±0.2** |
| 200 | 0.2 % | IR@1 | 1.1±0.1 | 1.2±0.1 | 1.2±0.1 | 1.7±0.1 | 2.4±0.1 | **3.8±0.1** |
| | | IR@5 | 4.6±0.3 | 4.7±0.1 | 5.1±0.2 | 6.5±0.4 | 9.3±0.2 | **13.4±0.1** |
| | | IR@10 | 8.3±0.6 | 8.5±0.2 | 8.9±0.2 | 12.3±0.8 | 15.5±0.2 | **21.8±0.2** |
| | | TR@1 | 1.7±0.2 | 1.6±0.2 | 1.9±0.1 | 3.3±0.2 | 4.3±0.1 | **5.3±0.2** |
| | | TR@5 | 6.5±0.5 | 6.6±0.2 | 6.7±0.2 | 11.9±0.6 | 14.2±0.3 | **17.3±0.2** |
| | | TR@10 | 11.1±0.6 | 11.2±0.4 | 11.6±0.3 | 19.4±1.2 | 22.6±0.2 | **27.0±0.2** |
| 500 | 0.4 % | IR@1 | 2.2±0.2 | 2.3±0.1 | 2.4±0.2 | 2.5±0.5 | 2.8±0.2 | **5.4±0.1** |
| | | IR@5 | 8.8±0.6 | 8.8±0.1 | 9.0±0.3 | 8.9±0.7 | 9.9±0.5 | **18.0±0.1** |
| | | IR@10 | 14.9±0.8 | 14.8±0.2 | 15.4±0.4 | 15.8±1.5 | 16.5±0.7 | **28.2±0.1** |
| | | TR@1 | 3.5±0.4 | 2.9±0.2 | 3.6±0.2 | 5.0±0.4 | 5.3±0.5 | **8.1±0.3** |
| | | TR@5 | 11.9±0.7 | 11.2±0.4 | 12.4±0.3 | 17.2±1.3 | 18.3±1.5 | **23.5±0.3** |
| | | TR@10 | 19.2±0.5 | 18.9±0.3 | 20.0±0.5 | 26.0±1.9 | 27.9±1.4 | **34.6±0.6** |

Table 16: Cross-architecture evaluation results on Flickr30k with $N = 100$ synthetic pairs.

| Method | Text Encoder | Image Encoder | IR@1 | IR@5 | IR@10 | TR@1 | TR@5 | TR@10 |
|---|---|---|---|---|---|---|---|---|
| Random | BERT | NFNet | 2.1 | 7.4 | 12.2 | 3.6 | 9.6 | 15.6 |
| | | NF-ResNet | 2.2 | 8.0 | 13.4 | 3.0 | 10.1 | 16.4 |
| | | NF-RegNet | 2.2 | 7.6 | 12.4 | 2.6 | 9.7 | 15.3 |
| | | ViT | 2.8 | 9.3 | 15.1 | 4.2 | 13.3 | 19.8 |
| | DistilBERT | NFNet | 2.4 | 8.6 | 14.4 | 3.2 | 11.1 | 16.4 |
| | | NF-ResNet | 2.6 | 9.6 | 15.3 | 3.6 | 11.4 | 18.4 |
| | | NF-RegNet | 2.3 | 8.0 | 13.4 | 2.9 | 10.0 | 15.4 |
| | | ViT | 2.9 | 10.6 | 17.4 | 4.1 | 13.1 | 20.6 |
| MTT-VL | BERT | NFNet | 4.7 | 15.7 | 24.6 | 9.9 | 28.3 | 39.1 |
| | | NF-ResNet | 1.8 | 7.0 | 11.9 | 2.9 | 10.5 | 16.4 |
| | | NF-RegNet | 1.5 | 6.0 | 10.6 | 2.9 | 9.2 | 15.0 |
| | | ViT | 2.1 | 7.8 | 13.1 | 3.9 | 12.2 | 18.6 |
| | DistilBERT | NFNet | 4.2 | 14.4 | 22.5 | 10.2 | 29.0 | 40.8 |
| | | NF-ResNet | 1.2 | 5.0 | 9.0 | 3.2 | 10.1 | 16.7 |
| | | NF-RegNet | 0.8 | 3.7 | 7.1 | 3.1 | 10.2 | 17.0 |
| | | ViT | 1.7 | 6.4 | 11.0 | 3.2 | 10.9 | 17.7 |
| LoRS | BERT | NFNet | 7.4 | 22.8 | 34.0 | 14.2 | 37.9 | 52.2 |
| | | NF-ResNet | 1.6 | 7.0 | 12.1 | 2.9 | 11.1 | 18.0 |
| | | NF-RegNet | 1.6 | 6.4 | 11.2 | 2.5 | 11.0 | 17.7 |
| | | ViT | 2.0 | 7.3 | 12.7 | 3.2 | 11.7 | 19.0 |
| | DistilBERT | NFNet | 5.2 | 17.3 | 27.2 | 12.5 | 34.0 | 45.0 |
| | | NF-ResNet | 1.7 | 6.3 | 10.8 | 4.0 | 11.9 | 18.8 |
| | | NF-RegNet | 1.3 | 5.2 | 8.9 | 3.6 | 12.0 | 18.6 |
| | | ViT | 1.4 | 5.8 | 9.9 | 3.2 | 12.7 | 20.4 |
| CovMatch | BERT | NFNet | 10.1 | 28.9 | 41.0 | 13.6 | 36.9 | 50.6 |
| | | NF-ResNet | 4.7 | 15.4 | 23.4 | 5.5 | 17.3 | 26.6 |
| | | NF-RegNet | 3.9 | 13.8 | 21.7 | 5.6 | 16.8 | 26.0 |
| | | ViT | 3.7 | 13.5 | 21.7 | 5.2 | 18.6 | 28.0 |
| | DistilBERT | NFNet | 8.5 | 25.1 | 36.7 | 12.6 | 33.4 | 46.2 |
| | | NF-ResNet | 5.3 | 16.2 | 25.1 | 6.2 | 17.6 | 26.4 |
| | | NF-RegNet | 3.8 | 13.4 | 21.9 | 5.3 | 17.7 | 25.6 |
| | | ViT | 3.0 | 11.9 | 19.5 | 5.0 | 16.3 | 24.6 |

