# OpenReview forum: "CovMatch: Cross-Covariance Guided Multimodal Dataset Distillation with Trainable Text Encoder"
_NeurIPS.cc/2025/Conference — NeurIPS 2025 poster_

### Official Review · Reviewer_cGeV · 2025-06-23

**Clarity:** 3
**Significance:** 2
**Originality:** 3
**Rating:** 4
**Confidence:** 3

**Summary:**

This paper studies dataset distillation problem aiming to distill a smaller dataset from a large dataset to achieve the best performance,  for vision-language model training. The authors analyze the bottlenecks of existing bi-trajectory matching methods. They further propose a new method to involve text encoder training in the distillation framework. To reduce computational overhead, they propose cross-covariance matching loss with a feature matching regularization. The experiments show that the proposed framework beats previous methods on two large datasets.

**Questions:**

In the weaknesses, I list a bunch of questions about the method and experiments. The authors just need to respond to my questions listed. I’ll also refer to the comments of other reviewers to decide final ratings.

**Ethical Concerns:**

["NO or VERY MINOR ethics concerns only"]

**Final Justification:**

After reading the rebuttal and the response to other reviewers, my concerns have been well addressed. I tend to keep my positive rating for this submission. I hope the author can futher polish this paper by involving the experiments and discussions in rebuttal.

**Limitations:**

Yes

**Quality:**

3

**Strengths And Weaknesses:**

Strengths:
1. This paper includes thorough analysis on widely-used bi-trajectory macing method, which shows more insights into the deficiency of previous methods.
2. This paper shows evidence for the proposed method using equations and experiments. 3. The improvement over previous baselines is prominent on the two datasets.

Weaknesses:
1. I don’t quite understand how to update the dataset during training. This paper says the goal is to synthesize a smaller dataset. How to under the word “synthesize” here? Selecting a subset of samples from the large dataset or generating completely new samples? How to backpropagate the gradient to update the dataset?
2. What’s the input embedding layer in Fig.1? Is it the encoding layer to map each word to a vector? Why not involve it in gradient backpropagation either?
3. In Eq.10, the expectation is over the whole real and synthetic datasetes. However, the loss is calculated based on batches during training. What’s the impact of training batch size to the final performance?
4. What’s upper bound of the proposed method by increasing the scale of the distilled datasets? It will show more insights if the authors can show the saturation point. 5. What’s the performance of train the VLM on entire Flickr30K and COCO datasets? I’m wondering the current gap between training using the distilled dataset and training using the full set.

---

> ### Author Rebuttal · Authors · 2025-07-31
>
> # Response to Reviewer cGeV
>
> We sincerely appreciate the constructive feedback provided by the reviewer. We have made every effort to address the concerns and questions raised. We hope that this response addresses the reviewer's questions.
>
> >**1. I don’t quite understand how to update the dataset during training. This paper says the goal is to synthesize a smaller dataset. How to understand the word “synthesize” here? Selecting a subset of samples from the large dataset or generating completely new samples? How to backpropagate the gradient to update the dataset?**
>
> In our work, the term “synthesize” refers to learning a set of synthetic image-text pairs, which are initialized from a randomly selected subset of real samples in the original dataset. These synthetic samples are parameterized as learnable tensors and are directly optimized through gradient-based methods.
>
> Specifically, each synthetic image and text sample is represented as a learnable tensor with shapes `[3,H,W]` and `[L,D]`, respectively, where L is the sequence length and D is the hidden dimension. During training, we compute the cross-covariance and feature-matching loss (as defined in Equation (11)) using these synthetic pairs. Since the synthetic samples are directly involved in the loss computation, we can backpropagate the gradient of the loss with respect to the synthetic inputs. This allows us to update the synthetic dataset $\mathcal{S}$ as follows:
> $$ \mathcal{S} \leftarrow \mathcal{S} - \eta \nabla_\mathcal{S} \mathcal{L}^\textrm{covmatch}(\mathcal{S}), $$
> where $\eta$ is the learning rate.
>
>
> >**2. What’s the input embedding layer in Fig.1? Is it the encoding layer to map each word to a vector? Why not involve it in gradient backpropagation either?**
>
> We thank the reviewer for pointing this out. Let us first clarify the process of text encoding in our framework. The input text, which is a sequence of words, is first tokenized into a sequence of token IDs using a tokenizer. These token IDs are then converted into dense vector representations by summing three types of embeddings: token embeddings, segment embeddings, and positional embeddings. The resulting embedding sequence is then fed into the transformer layers of the text encoder.
>
> In Figure 1, we refer to this entire process—i.e., from raw text to the input of the transformer layers—as the input embedding layer. We do not involve this part in gradient backpropagation because the input tokens are discrete and non-differentiable. Since gradients cannot be propagated through discrete inputs, we instead directly optimize continuous synthetic embeddings that bypass the input embedding layer, as explained in our method.
>
> We will clarify this terminology in the revised manuscript to avoid any confusion.
>
>
> >**3. In Eq.10, the expectation is over the whole real and synthetic datasetes. However, the loss is calculated based on batches during training. What’s the impact of training batch size to the final performance?**
>
> We thank the reviewer for the thoughtful question. While our original objective aims to match the cross-covariance matrix and mean feature of the entire training dataset with those of the synthetic set, computing the exact statistics over the full training set at every distillation step is computationally prohibitive—especially since the embeddings are continuously updated by the online model. To address this, we approximate the full-dataset statistics using mini-batches sampled from the training set at each step. This strategy introduces some variance, but allows the method to remain tractable and scalable during training.
>
> As shown in Table R1, increasing the training batch size leads to consistently improved retrieval performance. This trend suggests that larger batches enable more accurate approximation of the full-dataset statistics, particularly the cross-covariance between real and synthetic features, thereby facilitating more effective learning.  However, this performance gain comes at the cost of increased computational overhead. Specifically, we observe that the per-iteration time (sec/iter) rises substantially with larger batch sizes. This highlights a fundamental trade-off: while larger batches provide better statistical approximations and improved final accuracy, they require significantly more compute resources.
>
> We will include this analysis and discussion in the revised version of the paper.
>
>
> Table R1. Effect of training batch size on performance and distillation speed on Flickr30k with N=100.
> | Train batch size | 64   | 128  | 256  | 512  | 1024 |
> |------------------|------|------|------|------|------|
> | Performance      | 29.1 | 30.5 | 30.4 | 30.9 | 31.1 |
> | sec/iter         | 1.02 | 1.22 | 1.36 | 1.79 | 3.50 |
>
>
> >**4. What’s upper bound of the proposed method by increasing the scale of the distilled datasets? It will show more insights if the authors can show the saturation point.**
>
>
> We thank the reviewer for the helpful suggestion. We agree that analyzing the scaling behavior of distilled dataset size can provide further insight into the effectiveness and limitations of our method.
>
> To investigate this, we conducted additional experiments on the Flickr30K dataset by increasing the number of distilled image-text pairs N beyond 500. As shown in Table R2, our method continues to improve as N increases, and reaches a saturation point around **N = 2000**, where it closely matches the performance of training with randomly selected real samples of the same size.
>
> In contrast, we observe that baseline methods such as MTT-VL[1] and LoRS[2] tend to saturate at smaller scales, typically around N=500, and in some cases even exhibit performance degradation as N increases further. We note that we conducted extensive hyperparameter tuning for these baselines at each scale, including learning rates, maximum start epoch, and similarity rank. Despite these efforts, the observed degradation suggests that the optimization processes of the baselines are inherently less effective on large number of synthetic pairs.
>
> These results highlight the **strong scalability** of our method compared to existing approaches. We will include this new analysis, along with saturation curves, in the revised version of the paper.
>
> Table R2. Scalability comparison on Flickr30K.
> | Method   | N=100 | N=200 | N=500 | N=1000 | N=1500 | N=2000 |
> |----------|-------|-------|-------|--------|--------|--------|
> | Random   |  8.6  | 13.1  | 22.6  | 31.7   | 38.2   | 42.1   |
> | MTT-VL   | 20.4  | 21.2  | 25.0  | 19.0   | 18.2   | 18.2   |
> | LoRS     | 27.4  | 29.5  | 31.6  | 31.3   | 29.6   | 27.3   |
> | CovMatch | 30.5  | 34.4  | 38.4  | 41.2   | 41.8   | 43.7   |
>
> [1] Wu X, Zhang B, Deng Z, et al. Vision-language dataset distillation. TMLR 2024.
>
> [2] Xu Y, Lin Z, Qiu Y, et al. Low-Rank Similarity Mining for Multimodal Dataset Distillation. ICLR 2024.
>
>
> >**5. What’s the performance of train the VLM on entire Flickr30K and COCO datasets? I’m wondering the current gap between training using the distilled dataset and training using the full set.**
>
> We thank the reviewer for the question. We report the performance of full dataset and compare with the performance of our distilled dataset in Table 1. Note that the performance of full-dataset training is also reported in the caption of Table 2 in the main paper. As shown in Table R3, our distilled dataset with 2,000 samples (less than 7% of the original size) achieves around 60% of the full-dataset performance on average across metrics. Notably, for the TR@10 metric, our method achieves approximately 75% of the full-data performance. This highlights the strong scalability and data efficiency of our proposed method.
>
> We believe that these results demonstrate the ability of our approach to significantly reduce the reliance on large-scale annotated datasets. That said, a non-negligible performance gap still remains, and we consider closing this gap through improved distillation objectives and architectures to be an important direction for future work.
>
> Table R3. Comparison of the CovMatch's performance with full dataset on Flickr30k.
> | # Synthetic pairs | IR@1 | IR@5 | IR@10 | TR@1 | TR@5 | TR@10 | Avg  |
> |--------------|------|------|-------|------|------|-------|------|
> | 100     | 10.1 | 28.6 | 40.9  | 14.8 | 38.0 | 50.6  | 30.5 |
> | 500      | 14.7 | 38.4 | 51.4  | 19.9 | 46.7 | 59.5  | 38.4 |
> | 1000     | 16.3 | 41.0 | 53.2  | 22.3 | 51.4 | 63.3  | 41.2 |
> | 2000     | 18.7 | 44.7 | 56.5  | 25.7 | 54.2 | 68.2  | 44.7 |
> | Full set     | 48.7 | 79.2 | 87.2  | 61.6 | 85.9 | 91.5  | 75.7 |

---

> > ### Comment · Reviewer_cGeV · 2025-08-06
> > **Response of Reviewer**
> >
> > Thanks for the detailed explanation and the thorough response. My questions are well answered. These extra experiments should be added to the paper to make it more convincing. I will keep my positive decision for this submission.

---

> > > ### Author Response · Authors · 2025-08-06
> > >
> > > Deer Reviewer cGeV,
> > >
> > > We sincerely appreciate your insightful suggestions and constructive feedback, as well as the time and effort you dedicated to reviewing our manuscript. Your comments have been invaluable in improving the quality of our work. We will incorporate the additional experimental results and clarifications into the revised version of the paper.
> > >
> > > Sincerely,
> > >
> > > The authors

---

### Official Review · Reviewer_29i1 · 2025-06-30

**Clarity:** 3
**Significance:** 2
**Originality:** 2
**Rating:** 4
**Confidence:** 3

**Summary:**

The paper introduces CovMatch, a framework for multimodal dataset distillation that addresses the key challenge of cross-modal alignment in vision-language models. The core innovations include:
- A cross-covariance matching loss that aligns the cross-covariance matrices of real and synthetic data, enabling joint optimization of image and text encoders.
- A feature matching regularizer to ensure distribution consistency within each modality.
- An online model update strategy to prevent overfitting during distillation.
Experimental results on Flickr30K and COCO demonstrate that CovMatch outperforms state-of-the-art methods, while reducing computational costs significantly.

**Questions:**

See Weakness Part. If the concerns can be resolved, I will reconsider the final rating.

**Ethical Concerns:**

["NO or VERY MINOR ethics concerns only"]

**Final Justification:**

The additional theoretical analysis and more experiments on more modalities resolve my main concerns, I raise my score. I also hope the theoretical analysis given in rebutal will be inporporated into the final version of paper if accepted.

**Limitations:**

Yes

**Paper Formatting Concerns:**

No Paper Formatting Concerns

**Quality:**

3

**Strengths And Weaknesses:**

Strengths

- Novel Approach to Multimodal Distillation: By relaxing the assumption of frozen text encoders, CovMatch addresses a critical bottleneck in prior work. The cross-covariance alignment theoretically grounds the method in capturing semantic dependencies between modalities.
- Comprehensive Evaluation: The paper validates CovMatch across multiple datasets, metrics (IR@K, TR@K), and model architectures (NFNet, ViT, BERT, DistilBERT). Ablation studies on scaling factors, feature matching weights, and update strategies provide clear insights into the method’s effectiveness.

Weaknesses

- Theoretical Depth: While the cross-covariance alignment is motivated by linear contrastive learning, the paper lacks a formal proof of identifiability or convergence guarantees. For instance, it is unclear how the method handles non-linear feature spaces or complex causal structures in latent variables.

-  Limited Modality Scope: The current framework focuses on image-text pairs, ignoring other multimodal scenarios (e.g., video-text, audio-visual). Extending CovMatch to these domains would strengthen its generality.

- Insufficient comparison methods: Only two dataset distillation methods are used for performance comparison, and only one is published in 2024.

- Insufficient related works: For Multimodal Distillation, it is weird no studies for image-text retrieval are mentioned, such as [1-4].

[1] Karpathy, Andrej, and Li Fei-Fei. "Deep visual-semantic alignments for generating image descriptions." Proceedings of the IEEE conference on computer vision and pattern recognition. 2015.

[2] Li, Kunpeng, et al. "Visual semantic reasoning for image-text matching." Proceedings of the IEEE/CVF international conference on computer vision. 2019.

[3] Liu, Chunxiao, et al. "Graph structured network for image-text matching." Proceedings of the IEEE/CVF conference on computer vision and pattern recognition. 2020.

[4] Wang, Haoran, et al. "Coder: Coupled diversity-sensitive momentum contrastive learning for image-text retrieval." European Conference on Computer Vision. Cham: Springer Nature Switzerland, 2022.

---

> ### Author Rebuttal · Authors · 2025-07-31
>
> # Response to Reviewer 29i1
>
> We sincerely appreciate the constructive feedback provided by the reviewer. We have made every effort to address the concerns and questions raised. We hope that this response addresses the reviewer's questions.
>
> >**1. Theoretical Depth: While the cross-covariance alignment is motivated by linear contrastive learning, the paper lacks a formal proof of identifiability or convergence guarantees. For instance, it is unclear how the method handles non-linear feature spaces or complex causal structures in latent variables.**
>
> Thank you for this valuable comment. As the reviewer noted, our formulation of multimodal dataset distillation as a cross-covariance alignment problem is grounded in a linearized contrastive learning framework. Specifically, at each distillation iteration, we fix the image and text encoders and consider training the final linear projection layers to convergence. To mitigate overfitting to a static model state, we incorporate an online model update step: before each distillation step, the encoders are updated using a small batch of real data via a single gradient step.
>
> On the theoretical side, our linearized formulation enables a closed-form solution for the inner optimization. The optimal projection layers $G\_v$ and $G\_l$ minimize the following objective,
> $$-Tr(G\_v C^\mathcal{S} G\_l^\top ) + \frac{\rho}{2} \||G\_v^\top G\_l\||\_F^2 = \frac{\rho}{2}\||G\_v^\top G\_l - \frac{1}{\rho} C^\mathcal{S} \||\_F^2 - \frac{1}{2\rho}\||C^\mathcal{S}\||\_F^2,$$
> which admits a closed-form solution:
> $$\hat{G}\_v^\top \hat{G}\_l = \frac{1}{\rho} C^\mathcal{S}.$$
> This provides identifiability and convergence guarantees for the projection layers under the linear setting.
>
> While our theoretical formulation assumes linear representations—i.e., $z\_v=G\_vh\_v$ and $z\_l=G\_l h_l$, with  $h\_v=f\_v(x\_v;\theta\_v)$ and $h\_l=f\_l(x\_l;\theta\_l)$ for fixed encoder parameters—the insights extend more broadly to non-linear settings. In particular, recent work has shown that fine-tuning large neural networks (e.g., foundation models) often operates in the Neural Tangent Kernel (NTK) regime [1], where the training dynamics are well approximated by a linear model over a high-dimensional feature space derived from the model's gradients [2]. This connection suggests that our analysis in the linear regime can meaningfully inform behavior in more general, non-linear settings.
>
> Empirically, our experiments in Section 4 confirm that this formulation generalizes well: CovMatch, which is designed based on the linearized analysis, achieves state-of-the-art performance even when used with non-linear, high-capacity encoders like NFNet and BERT. The intuition is that synthetic data that effectively trains the linear projection layer tends to also provide strong training signals for the full network, particularly when the network is decomposed into a feature extractor and a task-specific head.
>
> We will incorporate these theoretical insights and supporting references in the revised manuscript.
>
> [1] Jacot, A., Gabriel, F., and Hongler, C. Neural tangent kernel: Convergence and generalization in neural networks. NIPS 2018.
>
> [2] Malladi, S., Wettig, A., Yu, D., Chen, D., and Arora, S. A kernel-based view of language model fine-tuning. ICML 2023.
>
>
> >**2. Limited Modality Scope: The current framework focuses on image-text pairs, ignoring other multimodal scenarios (e.g., video-text, audio-visual). Extending CovMatch to these domains would strengthen its generality.**
>
> We thank the reviewer for this insightful suggestion. To investigate the generality of CovMatch beyond image-text settings, we conducted additional experiments on the video-text retrieval task. We used a pretrained Video ResNet (r3d_18) [3] as the video encoder and BERT as the text encoder. Due to the time constraint, we constructed a reduced version of the WebVid-10M dataset [4], consisting of 49K training video-text pairs and 1K test pairs.
>
> As shown in Table R1, CovMatch consistently outperforms both MTT-VL [5] and LoRS [6] across nearly all evaluation metrics, demonstrating superior retrieval performance. These results indicate that CovMatch generalizes effectively to the video-text retrieval task, highlighting its strength as a robust and adaptable framework for multimodal dataset distillation across diverse modalities.
>
> Table R1. Performance comparison on video-text retrieval task.
> | Method    | VR@1 | VR@5 | VR@10 | TR@1 | TR@5 | TR@10 | Avg  |
> |-----------|------|------|-------|------|------|-------|------|
> | Random    | 1.1  | 2.7  | 5.2   | 0.7  | 3.3  | 5.0   | 3.0  |
> | MTT-VL    | 1.3  | 4.9  | 9.2   | 1.4  | 5.6  | 9.0   | 5.2  |
> | LoRS      | 1.2  | 5.4  | 9.3   | **1.9**  | 5.7  | 8.7   | 5.4  |
> | CovMatch  | **1.8**  | **7.6**  | **11.9**  | 1.8  | **6.9**  | **12.1**  | **7.0**  |
> | Full      | 15.2 | 38.9 | 53.6  | 14.2 | 38.4 | 52.8  | 35.5 |
>
> We will incorporate this new experiment and analysis into the revised version of the paper to highlight the potential applicability of our method to video-text retrieval. We thank the reviewer again for this valuable suggestion.
>
> [3] Tran, Du, et al. A closer look at spatiotemporal convolutions for action recognition. CVPR 2018.
>
> [4] Bain, Max, et al. Frozen in time: A joint video and image encoder for end-to-end retrieval. ICCV 2021.
>
> [5] Wu X, Zhang B, Deng Z, et al. Vision-language dataset distillation. TMLR 2024.
>
> [6] Xu Y, Lin Z, Qiu Y, et al. Low-Rank Similarity Mining for Multimodal Dataset Distillation. ICML 2024
>
>
> >**3. Insufficient comparison methods: Only two dataset distillation methods are used for performance comparison, and only one is published in 2024.**
>
> We thank the reviewer for the comment. We acknowledge the concern regarding the limited number of baseline methods used for comparison. However, we would like to clarify that multimodal dataset distillation is still a very nascent research area, and to the best of our knowledge, there currently exist only two publicly available methods that perform dataset distillation in the multimodal (specifically, vision-language) setting.
>
> It is worth noting that both existing baselines—MTT-VL [5] and LoRS [6]—are based on adaptations of the trajectory matching framework [7]. In principle, one could attempt to adapt other unimodal dataset distillation methods (e.g., DC[8], DM[9]) to the multimodal case. However, such extensions face significant challenges due to the importance of maintaining cross-modal alignment and the larger scale of multimodal models. As reported in [2], these adaptations often result in unstable optimization and performance degradation (see Table 1 in [6]).
> We agree that as the field evolves and more methods are introduced, future work should expand the range of comparisons accordingly. We will also make this limitation and justification more explicit in the revised version of the paper.
>
> [7] Cazenavette, George, et al. Dataset distillation by matching training trajectories. CVPR 2022.
>
> [8] Zhao, Bo, Konda Reddy Mopuri, and Hakan Bilen. Dataset Condensation with Gradient Matching. ICLR 2021.
>
> [9] Zhao, Bo, and Hakan Bilen. Dataset condensation with distribution matching. WACV 2023.
>
>
>
> >**4. Insufficient related works: For Multimodal Distillation, it is weird no studies for image-text retrieval are mentioned, such as [1-4].**
>
> We thank the reviewer for pointing out this important oversight. We agree that prior work on image-text retrieval and matching provides valuable context for our study, especially given that our method builds upon contrastive learning paradigms similar to those employed in retrieval systems.
>
> The works cited by the reviewer [10–13] have significantly advanced the field of image-text alignment and retrieval through various architectures such as visual-semantic embedding [10], semantic reasoning [11], graph-based matching [12], and contrastive momentum learning [13]. While these methods do not address dataset distillation directly, they are indeed relevant as they form the foundation for many modern image-text pretraining frameworks.
>
> To address this, we will add a new subsection titled “Image-Text Retrieval and Matching” under the Related Work section of our paper. In this subsection, we will briefly review these representative works and clarify how our setting differs—namely, in focusing on synthesizing compact datasets for training rather than improving retrieval performance directly.
>
> We appreciate the reviewer’s suggestion and will incorporate this discussion in the revised manuscript to provide a more comprehensive overview of related research.
>
> [10] Karpathy, Andrej, and Li Fei-Fei. Deep visual-semantic alignments for generating image descriptions. CVPR 2015.
>
> [11] Li, Kunpeng, et al. Visual semantic reasoning for image-text matching. ICCV 2019.
>
> [12] Liu, Chunxiao, et al. Graph structured network for image-text matching. CVPR 2020.
>
> [13] Wang, Haoran, et al. Coder: Coupled diversity-sensitive momentum contrastive learning for image-text retrieval. ECCV 2022.

---

> > ### Comment · Reviewer_29i1 · 2025-08-06
> > **Comment**
> >
> > Thanks for your detailed response. The additional theoretical analysis and more experiments on more modalities resolve my main concerns, I tend to raise my score.

---

> > > ### Author Response · Authors · 2025-08-06
> > >
> > > Dear Reviewer 29i1,
> > >
> > > Thank you again for taking the time to review our manuscript. We sincerely appreciate your constructive feedback, which helped us better articulate the strengths of our method. We're also very grateful for your quick response and for raising your score.
> > >
> > > Sincerely,
> > >
> > > The Authors

---

> ### Author Response · Authors · 2025-08-06
> **Gentle Reminder of the Discussion Deadline**
>
> Deer Reviewer 29i1,
>
> We sincerely appreciate your valuable suggestion and constructive feedback for improving our paper.
>
> For theoretical depth, we will incorporate the theoretical insights discussed in our response, along with supporting references, into the revised manuscript. Regarding the limited modality scope, we have extended our evaluation to the video-text domain, where our method demonstrates strong performance. With respect to the insufficient comparison, we understand your concern; however, to the best of our knowledge, only two baseline methods currently exist for this setting. In response to the point on insufficient related works, we will add a new subsection titled “Image-Text Retrieval and Matching” in the Related Work section of the revised manuscript.
>
> We understand your time is valuable and that you may have a busy schedule. As the discussion deadline is approaching, we kindly ask if you have any further suggestions or comments. Your feedback would be highly appreciated, and we look forward to hearing from you.
>
> Sincerely,
>
> The Authors

---

### Official Review · Reviewer_NCKk · 2025-07-02

**Clarity:** 4
**Significance:** 3
**Originality:** 2
**Rating:** 4
**Confidence:** 4

**Summary:**

This paper presents CovMatch, a multimodal dataset distillation method for image-text contrastive learning. The key improvements over existing approaches include: (1) avoiding trajectory storage and unrolling through cross-covariance matching, which provides a closed-form solution for the inner optimization; (2) enabling joint optimization of both image and text encoders rather than freezing the text encoder; and (3) achieving computational efficiency gains (1.2 sec/it vs 16.9 sec/it compared to prior methods). The approach reformulates the bi-level optimization as cross-covariance matrix alignment between real and synthetic features.

**Questions:**

## Questions

1. The paper claims computational efficiency by fixing encoders and only training projection layers to obtain a more tractable bi-level objective. However, given the extensive work in the VLM community on projection layer design and feature mapping [3], have the various hyperparameters, complexity considerations, and architectural choices for the projection layers been sufficiently explored and discussed?

2. This represents my main concern regarding the relationship with MTT-VL [1]. While the paper does compare with MTT-VL, I believe **the discussion of similarities and differences should be expanded in Section 2.2, rather than only discussing LORS.** The innovation appears somewhat limited relative to MTT-VL - both methods employ trainable projection layers after the two encoders. Although the InfoNCE linearization transformation is mathematically elegant, the paper would benefit from additional experiments or more insightful analysis to strengthen the theoretical contributions beyond what appears to be primarily a computational optimization trick of "fix encoders during distillation, then train them afterwards." The fundamental algorithmic differences from [1] beyond this computational efficiency gain need to be more clearly articulated with supporting experimental evidence.

3. Given the significant computational savings demonstrated by this method, could it be extended to accelerate video-text matching scenarios involving multiple video frames [4]? This exploration would be valuable and could potentially improve the evaluation of this work.

4. What are the conditions required for the linearization in lines 167-173? It appears that equation (5) holds under two conditions: "large τ / bounded inner product magnitudes for flatting softmax" and "zero-mean preprocessing of features." This needs to be clarified to meet NeurIPS community standards. I suggest placing the main conclusions in Section 3.1, with step-by-step derivations in the appendix to help readers from other fields follow along, as this work could potentially inspire other domains.

# minor revision
1. Add commas after formulas (if the following word is lowercase like "where") or periods (if the following word is capitalized)
2. Considering the method's effectiveness, Table 1 could be accompanied by a 2D plot to visualize performance with other baselines for more clear representation, as this method should be on the Pareto frontier.
3. The derivation in lines 167-173 **could be written in a more easy-to-follow manner,** particularly the transition to the cross-covariance matching objective -Tr(G_v C^D G_l^T) + (ρ/2)||G_v^T G_l||_F^2, and step-by-step derivations should be added in the appendix.

**Ethical Concerns:**

["NO or VERY MINOR ethics concerns only"]

**Final Justification:**

I've read the rebuttal and raised my score  accordingly

**Limitations:**

Yes

**Quality:**

3

**Strengths And Weaknesses:**

**Strengths:**

Multimodal dataset distillation is an important topic [1][2], and this work addresses a limitation of existing approaches by enabling text encoder optimization during distillation. The motivation is well-supported through convincing experimental analysis showing that frozen text encoders create semantic alignment bottlenecks, as evidenced by the t-SNE visualizations and intra/inter-similarity analyses in Figure 2. The theoretical derivation leading to the cross-covariance matching objective is mathematically sound and provides an elegant closed-form solution that significantly reduces computational overhead compared to trajectory matching methods.

**Weaknesses:**

While the paper does compare with MTT-VL [1], the innovation appears quite limited relative to this prior work. The theoretical foundations are not original - linear contrastive learning comes from prior work [2], and the importance of cross-covariance has already been recognized in multimodal learning. The core insight is essentially a computational trick: "fix encoders during distillation, then train them afterwards." This feels more like an engineering solution rather than an algorithmic breakthrough, and the fundamental differences from [1] beyond this computational optimization are not sufficiently substantial.

Additionally, given the extensive work in the VLM community on projection layer design and feature mapping [3], the paper would benefit from more thorough analysis of projection layer architectural choices, hyperparameters, and complexity considerations.

**References:**
[1] Wu X, Zhang B, Deng Z, et al. Vision-language dataset distillation[J]. TMLR, 2024.
[2] Xu Y, Lin Z, Qiu Y, et al. Low-Rank Similarity Mining for Multimodal Dataset Distillation[C]//International Conference on Machine Learning. PMLR, 2024: 55144-55161.
[3] Li J, Li D, Savarese S, et al. Blip-2: Bootstrapping language-image pre-training with frozen image encoders and large language models[C]//International conference on machine learning. PMLR, 2023: 19730-19742.
[4] https://huggingface.co/datasets/TempoFunk/webvid-10M

---

> ### Author Rebuttal · Authors · 2025-07-31
>
> # Response to Reviewer NCKk
>
> We sincerely appreciate the constructive feedback provided by the reviewer. Below, we addresses the reviewer's questions.
>
> >**1. While the paper does compare with MTT-VL [1], I believe the discussion of similarities and differences should be expanded in Section 2.2....The paper would benefit from a clearer articulation of the algorithmic differences from MTT-VL and stronger experimental or theoretical justification to support the claimed novelty.**
>
> We thank the reviewer for the comment and the opportunity to clarify the novelty of our method.
>
> First, we acknowledge that Section 2.2 primarily compares our method only with LoRS [2]. To address this more thoroughly, we have extended our analysis of the pairwise similarity (Fig. 2b) and scalability (Fig. 2c) plots to include MTT-VL. As shown in Table R1, CovMatch consistently exhibits a larger gap between intra- and inter-pair similarity across synthetic set sizes, compared to MTT-VL and LoRS. We also report additional scalability results with MTT-VL in Table R3 of the Reviewer xbh4 response, further supporting the strong scalability of CovMatch. These comparisons will be incorporated into the revised manuscript.
>
> Table R1. Extended comparison of gap between intra- and inter-pair average similarity.
> |Method|N=100|N=500|
> |-|-|-|
> |MTT-VL|0.387|0.284|
> |LoRS|0.389|0.378|
> |CovMatch|0.575|0.592|
>
> Second, we would like to highlight the key algorithmic distinctions that differentiate our method from MTT-VL.
> * **Distillation with large pretrained encoders:** A major limitation of MTT-VL is the difficulty to incorporate unfrozen large pretrained encoders such as BERT due to the high computational cost. In contrast, CovMatch introduces a lightweight cross-covariance matching objective, enabling efficient distillation with large, unfrozen encoders. The reviewer may be concerned that our method also assumes frozen encoders, which could diminish the distinction from prior work. However, we emphasize that while we assume frozen encoders to derive the cross-covariance matching objective, we do not fix them during distillation. Instead, we employ an online model update strategy to adapt the distilled set to evolving, unfrozen encoders.
> * **Distillation of text data:** Prior methods distill synthetic text that is directly fed into the text projection layer, which forces the text encoder to remain frozen. In contrast, our method distills inputs to the transformer layers of the text encoder, which maximizes the performance gain achievable through distillation.
> * **Cross-covariance matching algorithm:** We find that aligning the cross-covariance matrices between real and synthetic pairs is critical for effective multimodal distillation. However, as showin in Table R2, the baselines fail to successfully align the cross-covariance, and in fact, MTT-VL even increases the misalignment from initial state. In contrast, our method directly maximizes the alignment using cross-covariance matching loss term, resulting in state-of-the-art performance.
>
> Table R2. Comparison of the Frobenius distance between the cross-covariance matrices of the real and distilled sets, $||C^\mathcal{T}-C^\mathcal{S}||_F$.
> |Method| Frobenius distance |
> |-|-|
> |Random (initial)|53.9|
> |MTT-VL|157.6|
> |LoRS |57.1|
> |CovMatch| 7.8 |
>
> We introduce a scalable and principled framework for multimodal dataset distillation that departs from conventional practices in the field. We believe this work constitutes a meaningful advancement in the field, beyond simple computational trick.
>
> [1] Wu X, Zhang B, Deng Z, et al. Vision-language dataset distillation. TMLR 2024.
>
> [2] Xu Y, Lin Z, Qiu Y, et al. Low-Rank Similarity Mining for Multimodal Dataset Distillation. ICLR 2024.
>
>
> >**2. The paper claims computational efficiency by fixing encoders and training only projection layers for a tractable bi-level objective. But given extensive work on projection design in the VLM community [3], have the architectural choices and hyperparameter tuning been sufficiently explored?**
>
> We thank the reviewer for this insightful suggestion. In our implementation, we adopt a two-layer projection network with GELU activation, following prior baseline designs. While this setup has demonstrated strong performance, we agree that exploring the architectural design space of projection layers is a meaningful direction.
>
> To address this, we conducted ablation studies on two aspects: projection dimension (i.e., dimension of projected output) and projection depth (i.e., number of layers in the projection network). Note that our default configuration uses a projection dimension of 2304. As shown in Table R3, reducing the projection dimension (e.g., 768, 256) leads to noticable performance degradation, likely due to loss of information. Also, as shown in Table R4, single-layer projection network achieved performance comparable to our default two-layer setup, but deeper networks (3 layers) significantly degraded the performance. These results suggest that careful design of the projection network is essential for achieving strong performance.
>
> Table R3. Effect of projection dimension on Flickr30k with N=100.
> |Projection Dim|256|768|2304|4096|
> |-|-|-|-|-|
> |Performance (avg)|16.1|26.0|30.5|28.9|
>
> Table R4. Effect of projection depth on Flickr30k with N=100.
> |Projection Depth|1|2|3|
> |-|-|-|-|
> |Performance (avg)|29.5|30.5|19.9|
>
> In this context, the approach proposed in [3]—which freezes the encoder and instead trains a more expressive projection module, such as a transformer-based adapter (e.g., Q-former)—offers a promising alternative. We believe integrating such ideas into distillation setting, where they have yet to be systematically explored, could open up meaningful future direction. We thank the reviewer again for this valuable suggestion.
>
> We will incorporate these ablation results and the related discussion in the revised version.
>
> [3] Li J, Li D, Savarese S, et al. Blip-2: Bootstrapping language-image pre-training with frozen image encoders and large language models. ICML 2023.
>
> >**3. Given the significant computational savings demonstrated by this method, could it be extended to accelerate video-text matching scenarios involving multiple video frames [4]?**
>
> We thank the reviewer for this insightful suggestion. We conducted additional experiments on the video-text retrieval task using pretrained VideoResNet (r3d_18) [5] as the video encoder and BERT as the text encoder. Since it is computationally impractical to distill full WebVid-10M dataset [4] within our current time constraint, we instead used reduced dataset which consists of 49K training video-text pairs and 1K test pairs. For the synthetic set, we distilled 100 video-text pairs, using 8 sampled frames per video.
>
> As shown in Table R5, CovMatch outperforms both MTT-VL [1] and LoRS [2], demonstrating superior retrieval performance. The result indicates that CovMatch generalizes effectively to the video-text retrieval task, underscoring its robustness across modalities. Its strong performance on this more complex task also highlights the CovMatch’s computational efficiency.
>
> Table R5. Performance comparison on video-text retrieval task with N=100.
> |Method|Random|MTT-VL|LoRS|CovMatch|
> |-|-|-|-|-|
> |Performane (avg)|3.0|5.2|5.4|**7.0**|
>
> We will incorporate this new experiment and analysis into the revised version of the paper to highlight the potential applicability of our method to video-text retrieval.
>
> [4] Bain, Max, et al. Frozen in time: A joint video and image encoder for end-to-end retrieval. ICCV 2021.
>
> [5] Tran, Du, et al. A closer look at spatiotemporal convolutions for action recognition. CVPR 2018.
>
>
> >**4. What are the conditions required for the linearization in lines 167-173?**
>
> >**Minor Revision 3. The derivation for the transition to the cross-covariance matching objective $-Tr(G_v C^D G_l^T) + (\rho/2)||G_v^T G_l||_F^2$.**
>
> To clarify, in our main analysis, we do not impose any assumptions to directly linearize the InfoNCE loss. Instead, we adopt the linear contrastive loss defined in Eq. (5) as a standalone objective to simplify the bi-level optimization in multimodal distillation. This linear form enables a natural connection to the cross-covariance alignment term in Eq. (6), ultimately leading to the simplified trace-based objective in Eq. (8), $Tr((\mathcal{C}^{\mathcal{T}})^\top \mathcal{C}^{\mathcal{S}} )$.
>
> That said, as the reviewer correctly points out, the linear loss in Eq. (5) can also be derived as a high-temperature approximation of the InfoNCE loss.
> When $\tau$ is large, the softmax flattens, and a first-order Taylor expansion gives:
> $$\log(\sum_{j\neq i} \exp(s_{ij}/\tau))\approx \frac{1}{\tau(|\mathcal{D}|-1)}\sum_{j\neq i} s_{ij}+\log (|\mathcal{D}|-1).
> $$
> Substituting this into the InfoNCE loss (1) leads to the linearized form Eq. (5). This derivation provides further justification for using the linear objective in Eq. (5).
>
> Furthermore, the equivalence between Eq. (5) (the linear contrastive loss) and Eq. (6) (the cross-covariance matching objective) holds without any assumptions. The similarity terms in Eq. (5) can be expressed using the projected embeddings:
> $$s_{ij} := (G_v h_v^i)^\top (G_l h_l^j)=\sum_{k=1}^z (g_{v_k})^\top h_v^i ({h_l^j})^\top g_{l_k},$$
> where $G_v^\top=[g_{v_1},\dots, g_{v_z}]$ and $G_l^\top=[g_{l_1},\dots, g_{l_z}]$.
>
> The cross-covariance matrix $C^\mathcal{D}$ is defined as:
> $$
> C^{\mathcal{D}}:=\frac{1}{(|\mathcal{D}|-1)}\sum_{i=1}^{|\mathcal{D}|}(h_v^i-\mu_{h_v})(h_l^i-\mu_{h_l})^\top,
> $$
> and the term $-Tr(G_v C^{\mathcal{D}} G_l^\top)$ expands as
> $$
> -\sum_{k=1}^z g_{v_k}^\top  C^{\mathcal{D}}g_{l_k}.
> $$
> By aligning the summation forms, we establish the equivalence between the linear contrastive loss and cross-covariance matching objective.
> We will revise Section 3.1 for clarity and include the full derivation in the appendix.
>
> >**We will incorporate Minor Revision 1 and 2 accordingly.**

---

> ### Author Response · Authors · 2025-08-06
> **Gentle Reminder of the Discussion Deadline**
>
> Deer Reviewer NCKk,
>
> We sincerely appreciate your valuable suggestion and constructive feedback for improving our paper.
>
> In response, we clarified the novelty of our method with a more detailed comparison against MTT-VL, and we conducted additional experiments to explore the design space of projection layers. We also extended our evaluation to the video-text domain, where our method demonstrates strong performance. Additionally, we clarified that no specific conditions are required for the linearization in lines 167–173; however, as you correctly pointed out, the linear loss in Eq. (5) can indeed be derived as a high-temperature approximation of the InfoNCE loss. We will incorporate your thoughtful minor revision suggestions into the final version of the manuscript.
>
> We understand your time is valuable and that you may have a busy schedule. As the discussion deadline is approaching, we kindly ask if you have any further suggestions or comments. Your feedback would be highly appreciated, and we look forward to hearing from you.
>
> Sincerely,
>
> The Authors

---

> > ### Comment · Reviewer_NCKk · 2025-08-07
> >
> > The issue I wrote is well explained, therefore I'll raise my score by 1

---

> > > ### Author Response · Authors · 2025-08-07
> > >
> > > Dear Reviewer NCKk,
> > >
> > > Thank you once again for your time and thoughtful review of our manuscript. We sincerely appreciate your constructive feedback and suggestions, which helped us better articulate the strengths of our method. We're also very grateful for your quick response and for raising your score.
> > >
> > > Sincerely,
> > >
> > > The Authors

---

### Official Review · Reviewer_xbh4 · 2025-07-03

**Clarity:** 3
**Significance:** 3
**Originality:** 3
**Rating:** 4
**Confidence:** 5

**Summary:**

This paper introduces CovMatch, a novel framework for multimodal dataset distillation (MDD). The authors first unfreez the text encoder for learning efficacy. By analyzing the linearized model, CovMatch addresses the MDD problem by matching the covariance matrices between modalities. With additional online model updating technique, CovMatch achieves both significant efficiency and performance on multimodal retrieval tasks.

**Questions:**

1. The text encoder is unfrozen in only distillation stage or both distillation and network training (evaluation) stages? The latter one involves unfair comparison to baselines and the authors could unfreeze the text encoder of the baselines during evaluation.
2. There seems not to be a clear ablation comparison of CovMatch without unfrozen encoder (due to Q1). If the blue bars in Fig.5(c) are, their performance is comarable to baselines, which reduces the performance contribution of CovMatch itself.
3. In Sec.2.2, the authors discuss the limitation of MDD baselines at performance scaling (starts from line 142). Thus, a comparison at 1000 pairs would support the claim.

I would be glad to raise my rating if my concerns are addressed.

**Ethical Concerns:**

["NO or VERY MINOR ethics concerns only"]

**Final Justification:**

I appreciate the authors' feedback. My questions are sufficiently addressed, so I would keep my positive rating.

**Limitations:**

yes

**Quality:**

3

**Strengths And Weaknesses:**

## Strengths

- The paper is well-motivated. The proposed approach is reasonable, with both theoretical novelty and interesting technical contributions.
- The empirical evaluation is strong, exhibiting both distillation efficiency and good performance.
- The writing is clear.

## Weakness

The main weaknesses come from the ablation. The paper's central claim is that unfreezing the text encoder helps MDD and improves the performance when synthetic data scales up. This, however, could potentially lead to unfair comparison. Please see my "Questions" for details. Overall, I think the method is self-contained and addressing these concerns would enhance the clarity of the paper.

---

> ### Author Rebuttal · Authors · 2025-07-31
>
> # Response to Reviewer xbh4
>
> We sincerely appreciate the constructive feedback provided by the reviewer. We have made every effort to address the concerns and questions raised. We hope that this response addresses the reviewer's questions.
>
> >**1. The text encoder is unfrozen in only the distillation stage or both the distillation and network training (evaluation) stages? The latter one involves unfair comparison to baselines and the authors could unfreeze the text encoder of the baselines during evaluation.**
>
> We thank the reviewer for raising this important point. We unfreeze the text encoder during **both the distillation and evaluation stages** in our method. We understand the reviewer's concern regarding the fairness of comparison. However, it is infeasible to unfreeze and fine-tune the text encoder during the evaluation stage for existing baselines due to structural limitations in how they treat the distilled text.
>
> To elaborate, consider a baseline method that distills a synthetic dataset (e.g., 100 image-text pairs from Flickr). When training with the distilled set at the evaluataion stage, the image component (e.g., shape `[100, 3, 224, 224]`) is passed through the image encoder, while the distilled text component (e.g., `[100, 768]`) is directly used as input to the text projection layer, bypassing the text encoder entirely. As a result, the text encoder is never involved or updated during training. Once training is complete, the learned image encoder and projection layers are used in conjunction with the original frozen text encoder (e.g., pretrained BERT) to process real test samples.
>
> Hence, to ensure a fair comparison, we adapt our evaluation framework (Fig. 1) for the baseline methods. Specifically, instead of distilling text embeddings as direct inputs to the projection layer, we modify the baselines to newly distill synthetic embeddings that serve as inputs to the transformer layers within the text encoder. This allows the text encoder to be effectively trained on the synthetic set generated by the baselines. In this modified setup, we keep the text encoder frozen during distillation and unfreeze it only during the evaluation stage. This design choice is driven by two key challenges: (1) unfreezing the text encoder during distillation incurs substantial computational and storage overhead; and (2) despite the high cost, we attempted to unfreeze the text encoder and apply trajectory matching that includes it, but observed that trajectory matching loss for the text encoder fails to decrease, unlike the projection layers. This results in optimization failure and unstable training dynamics.
>
> As shown in Table R1, unfreezing the text encoder for evaluation improves baseline performance slightly, but our method still significantly outperforms them. This observation confirms that baseline methods, which do not incorporate the text encoder during distillation, cannot fully benefit from unfreezing it during evaluation, resulting in only marginal performance gains.
>
> Table R1. Results of unfreezing text encoder during the evaluation stage of baseline method on Flickr30k with N=500.
> | Method           | IR@1 | IR@5 | IR@10 | TR@1 | TR@5 | TR@10 | Avg  |
> |------------------|------|------|-------|------|------|-------|------|
> | LoRS [1] (frozen)    | 10.0 | 28.9 | 41.6  | 15.5 | 39.8 | 53.7  | 31.6 |
> | LoRS [1] (unfrozen)  | 10.0 | 28.0 | 41.5  | 17.7 | 42.5 | 57.0  | 32.8 |
> | CovMatch         | 14.7 | 38.4 | 51.4  | 19.9 | 46.7 | 59.5  | 38.4 |
>
> [1] Xu Y, Lin Z, Qiu Y, et al. Low-Rank Similarity Mining for Multimodal Dataset Distillation. ICLR 2024.
>
> >**2. There seems not to be a clear ablation comparison of CovMatch without unfrozen encoder (due to Q1). If the blue bars in Fig.5(c) are, their performance is comarable to baselines, which reduces the performance contribution of CovMatch itself.**
>
> Thank you for raising this important point. We would like to first clarify a possible misunderstanding regarding the interpretation of Figure 5(c). The term *“fixed”* in Figure 5(c) does not refer to freezing the text encoder during the evaluation stage, as discussed in Q1. Instead, it refers to whether the online model—i.e., the image and text encoders—is updated during the distillation process.
>
> To prevent the synthetic data from overfitting to a static encoder, our method introduces an online model update strategy during distillation: before each distillation step, the encoders are updated using a batch of real data. In Figure 5(c), the *blue bars* represent the setting where both the image and text encoders are fixed at their pretrained initialization throughout the distillation phase. In contrast, the *orange* or *green bars* represent settings where encoders are updated with syntehtic or real data during distillation. Importantly, in all three settings shown in Figure 5(c), the text encoder is unfrozen during the evaluation stage. Therefore, this figure should not be interpreted as an ablation on whether text encoder is frozen or not, but instead as an analysis of the effect of updating the online model during distillation.
>
> To more directly examine the role of the text encoder, we conducted an additional ablation study that investigates the effect of freezing it during both the distillation and evaluation stages. As shown in Table R2, freezing the text encoder during evaluation leads to a significant performance drop, highlighting the importance of training not only the image encoder and projection layers but also the text encoder for strong image-text retrieval ability. Moreover, we observe that freezing the text encoder during distillation also causes a substantial drop in performance (from 38.4 to 29.4 in average score). These findings suggest that involving the text encoder during distillation is critical for maximizing the effectiveness of our method.
>
> Table R2. Effect of freezing the text encoder during the distillation and evaluation stages of CovMatch on Flickr30k with N=500.
> | Distill  | Eval     | IR@1 | IR@5 | IR@10 | TR@1 | TR@5 | TR@10 | Avg  |
> |----------|----------|------|------|-------|------|------|-------|------|
> | Frozen   | Frozen   | 4.3  | 13.4 | 21.1  | 6.5  | 20.6 | 30.6  | 16.1 |
> | Frozen   | Unfrozen | 9.8  | 27.8 | 40.2  | 15.1 | 35.8 | 48.0  | 29.4 |
> | Unfrozen | Frozen   | 4.5  | 14.7 | 22.6  | 8.4  | 23.2 | 33.4  | 17.8 |
> | Unfrozen | Unfrozen | 14.7 | 38.4 | 51.4  | 19.9 | 46.7 | 59.5  | 38.4 |
>
> For a fair comparison where text encoder is unfrozen in both our method and baselines, we refer the reviewer to **Table R1 in Response 1**, which directly addresses this point. We will include this ablation study in the revised version to support a fair comparison with existing baselines.
>
>
> >**3. In Sec.2.2, the authors discuss the limitation of MDD baselines at performance scaling (starts from line 142). Thus, a comparison at 1000 pairs would support the claim.**
>
> We thank the reviewer for this helpful suggestion. We agree that comparison at more than 500 pairs is required to support our strong ability at performance scaling. As shown in Table R3, our method continues to improve as N increases until N=2000. In contrast, baseline methods such as MTT-VL [2] and LoRS [1] tend to saturate at smaller scales, typically around N=500, and in some cases even exhibit performance degradation as N increases further. We note that we conducted extensive hyperparameter tuning for these baselines at each scale, including learning rates, maximum start epoch, and similarity rank. Despite these efforts, the observed degradation suggests that the optimization processes of the baselines are inherently less effective on large number of synthetic pairs.
>
> These results highlight the strong scalability of our method compared to existing approaches. We will include this extended comparison, along with scaling curves, in the revised version of the paper.
>
> Table R3. Scalability comparison on Flickr30K.
> | Method   | N=100 | N=200 | N=500 | N=1000 | N=1500 | N=2000 |
> |----------|-------|-------|-------|--------|--------|--------|
> | Random   |  8.6  | 13.1  | 22.6  | 31.7   | 38.2   | 42.1   |
> | MTT-VL   | 20.4  | 21.2  | 25.0  | 19.0   | 18.2   | 18.2   |
> | LoRS     | 27.4  | 29.5  | 31.6  | 31.3   | 29.6   | 27.3   |
> | CovMatch | 30.5  | 34.4  | 38.4  | 41.2   | 41.8   | 43.7   |
>
> [2] Wu X, Zhang B, Deng Z, et al. Vision-language dataset distillation. TMLR 2024.

---

> > ### Comment · Reviewer_xbh4 · 2025-08-05
> >
> > I appreciate the authors' feedback. My questions are sufficiently addressed, so I would hold a positive rating.

---

> > > ### Author Response · Authors · 2025-08-06
> > >
> > > Deer Reviewer xbh4,
> > >
> > > We sincerely appreciate your valuable suggestion and constructive feedback for improving our paper. Moreover, we are truly grateful for the time and consideration you have invested in reviewing our manuscript. We will incorporate the additional experimental results and clarifications into the revised version of the paper.
> > >
> > > Sincerely,
> > >
> > > The authors

---

### Note · Authors · 2025-08-13

We would like to express our sincere gratitude to all reviewers and the Area Chair for their time, effort, and constructive feedback, which have greatly helped us to improve the quality and clarity of our work. Below, we first concisely restate the key contributions of our paper, followed by the additional analyses and clarifications made in response to the reviewers’ comments.

# Contributions
- We identify that distillation with a frozen text encoder results in limited semantic alignment, leading to poor performance scalability.
- We formulate multimodal dataset distillation as a cross-covariance matching objective, derived from an analysis of the linearized setup.
- The computational efficiency of our method enables the use of large encoder models (e.g., BERT) during distillation.
- We achieve state-of-the-art performance on Flickr30k and COCO datasets across varying number of synthetic pairs $N$, and exhibit remarkably strong cross-architecture generalization ability and performance scalability.

# Additional Analyses
- We extend our method to the video–text domain, with results further demonstrating its superiority (Reviewers NCKk and 29i1).
- For a fairer comparison, we adapt our framework to baseline methods to allow unfreezing the text encoder at the evaluation stage, yet our method still achieves superior performance (Reviewer xbh4).
- We conduct an ablation study on unfreezing the text encoder during both distillation and evaluation (Reviewer xbh4).
- We extend our experiments to $N=2000$, further demonstrating strong scalability (Reviewers xbh4 and cGeV).
- We explore a broader design space for the projection layer (Reviewer NCKk).
- We investigate the effect of training batch size (Reviewer cGeV).
- We include MTT-VL in our analysis (Section 2.2) and highlight key innovations of our approach compared to MTT-VL (Reviewer NCKk).

# Additional Clarifications
- We explicitly clarify how we learn synthetic data, especially textual data (Reviewer cGeV).
- We address concerns regarding the sufficiency of baseline comparisons (Reviewer 29i1).
- We provide more explicit theoretical insights and justifications for the cross-covariance alignment objective within the linearization framework (Reviewer 29i1).
- We offer a clearer explanation of the linearization of the InfoNCE loss and address concerns regarding the required assumptions or conditions (Reviewer NCKk).
- We extend the related work section to include image–text retrieval tasks (Reviewer 29i1).

---

### Decision · Program_Chairs · 2025-09-17

**Decision:**

Accept (poster)

**Comment:**

This paper received ratings (4, 4, 4, 4). It introduces CovMatch a framework for multimodal dataset distillation. All reviewers agree on acceptance. The reviewers find it well-motivated and technically sound about the method of unfreezing the text encoder and matching covariance matrices between modalities. Some initial questions raised regarding the experimental comparison and scalability were addressed during the rebuttal. The AC panel recommends accept.